# How Low Can You Go? Searching for the Intrinsic Dimensionality of Complex Networks using Metric Node Embeddings

**Nikolaos Nakis**[1]* **Niels Raunkjær Holm**[2]* **Andreas Lyhne Fiehn**[2]* **Morten Mørup**[2]
[1]Yale University   [2]Technical University of Denmark
`nicolaos.nakis@gmail.com, nielsraunkjaer@gmail.com,`
`andreas@lyhnefiehn.dk, mmor@dtu.dk`

## Abstract

Low-dimensional embeddings are essential for machine learning tasks involving graphs, such as node classification, link prediction, community detection, network visualization, and network compression. Although recent studies have identified exact low-dimensional embeddings, the limits of the required embedding dimensions remain unclear. We presently prove that lower dimensional embeddings are possible when using Euclidean metric embeddings as opposed to vector-based Logistic PCA (LPCA) embeddings. In particular, we provide an efficient logarithmic search procedure for identifying the exact embedding dimension and demonstrate how metric embeddings enable inference of the exact embedding dimensions of large-scale networks by exploiting that the metric properties can be used to provide linearithmic scaling. Empirically, we show that our approach extracts substantially lower dimensional representations of networks than previously reported for small-sized networks. For the first time, we demonstrate that even large-scale networks can be effectively embedded in very low-dimensional spaces, and provide examples of scalable, exact reconstruction for graphs with up to a million nodes. Our approach highlights that the intrinsic dimensionality of networks is substantially lower than previously reported and provides a computationally efficient assessment of the exact embedding dimension also of large-scale networks. The surprisingly low dimensional representations achieved demonstrate that networks in general can be losslessly represented using very low dimensional feature spaces, which can be used to guide existing network analysis tasks from community detection and node classification to structure revealing exact network visualizations.

*Code available at: https://github.com/AndreasLF/HowLowCanYouGo.*

## 1 Introduction

Graphs are used in a plethora of settings to model various complex systems including social networks, the Internet, citation links between research publications, neural networks, protein-protein interactions, food webs, and metabolic networks (Newman, 2003), to mention but a few. From a machine learning perspective graph representation learning (GRL) aiming to embed graph structure using low-dimensional vector spaces that provide compressed representations of network structure has in recent years garnered substantial attention (Hamilton et al., 2017a;b). Some of the challenges in GRL include preserving structure from the discrete graph space in the learned embedding space. This means that the connectedness and similarity of nodes in the graph should carry over into the embedding space (Zhang et al., 2020). Both local connectivity and global community structures are crucial to the characteristics of the system modeled by the graph. The concept of homophily (Mcpherson et al., 2001) should thus be preserved, such that links between nodes are signified in their corresponding embeddings. Intuitively, the node representations should be close in proximity to each other in some measure relevant to the embedding space. Further, this notion of proximity should apply to next-step neighbors, next-next-step, and so on (Zhang et al., 2017). In the literature,

---

*Shared first authorship.

various so-called shallow embedding methods, which essentially refer to mapping nodes one-to-one to embedding vectors have been proposed. These methods each aim to capture different parts of the graph structure, e.g. DeepWalk (Perozzi et al., 2014), LINE (Tang et al., 2015), node2vec (Grover & Leskovec, 2016), GraRep (Cao et al., 2015), TADW (Serrano et al., 2007), and many more (Zhang et al., 2020). Recently, it has been demonstrated that latent space network representation approaches such as the latent distance model (Hoff et al., 2002) perform favorably using ultra-low dimensional (i.e., $D = 2$ and $D = 3$) embedding representations (Nakis et al., 2022).

In Seshadhri et al. (2020) the goal of embedding graphs was formulated as capturing as much structure as possible from the graph in a low-rank representation. They also pose the question of how well we can embed graphs. They conclude that graphs created from low-dimensional embeddings cannot have many triangles involving vertices of low-degree. This was later discussed in Chanpuriya et al. (2020) where it was demonstrated that with a relaxed version of the factorization model, finding exact low-rank embeddings for bounded degree graphs, i.e. graphs with at most degree $k_{max}$, is indeed feasible. For this task, they used a method for learning an embedding model based on logistic principal component analysis (LPCA). However, the work only provided rough estimates of the required embedding dimensions and the approach was restricted to the analysis of small networks (i.e., less than 20.000 nodes).

In this paper, we investigate the limits required to achieve exact network embeddings. Specifically, we propose an algorithm to efficiently search for an upper bound for the exact embedding dimension ($D^*$) of graphs. We further theoretically and empirically demonstrate that lower embedding dimensions can be achieved considering metric embeddings using the latent distance model (LDM) (Hoff et al., 2002). Importantly, through the properties of distance functions, metric spaces have natural and intuitive notions of proximity and by extension also node homophily and interconnectedness on both local and global scale. Inspired by the Hierachical Block Distance Model (HBDM), as introduced by Nakis et al. (2022), we exploit that the metric properties are especially useful for embedding very large graphs. As the reconstruction check itself becomes intractable for large sets of nodes, we further exploit how hierarchical representations of data with metric properties allow linearithmic runtime complexity. As such, we implement a KD-tree-based nearest neighbor reconstruction check method.

## 1.1 RELATED WORK

The interest in embedding networks using low dimensional representations has been substantially explored in the inexact reconstruction setting in which latent space modeling approaches including the latent eigenmodel and latent distance model have been explored (Hoff et al., 2002; Hoff, 2007). Recently, it has been observed that the latent distance model using ultra-low dimensional representation provides strong generalization in graph representation learning tasks such as node classification and link prediction (Nakis et al., 2022). The first attempt to quantify the intrinsic dimensionality of exact network reconstruction was considered in Chanpuriya et al. (2020) based on a Logistic PCA (LPCA) model. For a network of $N$ nodes, the LPCA model consists of two rank $D \in \mathbb{Z}^+$ embedding matrices $\mathbf{X}, \mathbf{Y} \in \mathbb{R}^{N \times D}$. It assumes that the probability of a link between node $i$ and $j$, expressed in the adjacency matrix $\mathbf{A} \in \{0, 1\}^{N \times N}$ as $a_{i,j} = 1$, is $\sigma([\mathbf{X}\mathbf{Y}^\top]_{i,j})$ where $\sigma(x) = (1 + e^{-x})^{-1}$ is the sigmoid-function. Using the shifted adjacency matrix $\tilde{a}_{i,j} = 2a_{i,j} - 1$, the log-likelihood can be compactly written as $\log(\sigma(\tilde{a}_{i,j}[\mathbf{X}\mathbf{Y}^\top]_{i,j}))$. Maximizing the likelihoods of links between all node indices $i$ and $j$ is then equivalent to the following optimization task:

$$\min_{\mathbf{X}, \mathbf{Y}} \mathcal{L}(\mathcal{R}_{LPCA}(\mathbf{X}, \mathbf{Y})) = \min_{\mathbf{X}, \mathbf{Y}} \sum_{i=1}^{N} \sum_{j=1}^{N} -\log \sigma \left( \tilde{a}_{i,j} \left[ \mathbf{X}\mathbf{Y}^T \right]_{i,j} \right). \tag{1}$$

Notably, this objective is closely related to identifying the sign-rank of a matrix in which the product $\tilde{a}_{i,j} \left[ \mathbf{X}\mathbf{Y}^T \right]_{i,j} > 0 \ \forall i, j$. It has been shown that the sign-rank is lower bounded by $N/\sigma_{max}(\tilde{\mathbf{A}})$ where $\sigma_{max}(\tilde{\mathbf{A}})$ denotes the largest singular value, i.e. spectral norm, of $\tilde{\mathbf{A}}$ (Forster, 2002). This bound has been refined in the context of learning theory and communication complexity in (Razborov & Sherstov, 2010) whereas the existing known lower bounds for sign-rank have recently been surveyed in (Hatami et al., 2022) and found to have limitations. To minimize $\mathcal{L}(\mathcal{R}_{LPCA}(\mathbf{X}, \mathbf{Y}))$ Chanpuriya et al. (2020) initialize $\mathbf{X}$ and $\mathbf{Y}$ uniformly at random in $[-1, 1]$ and use the SciPy implementation of the L-BFGS scheme with default parameters and a maximum of 2000 iterations to optimize the

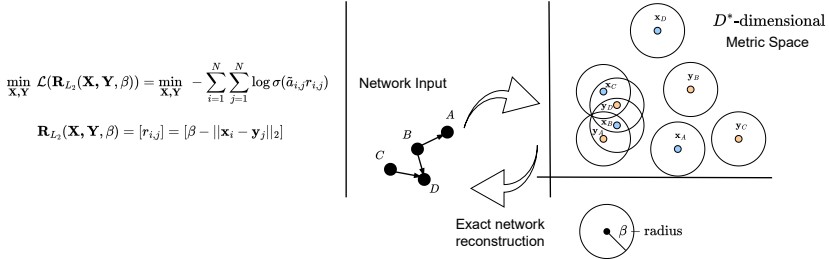

Figure 1: **Model Overview**: The input network is embedded into a low-dimensional space using matrices $\mathbf{X}$ and $\mathbf{Y}$, defining an upper bound $D^*$ on intrinsic dimensionality for structure-preserving reconstruction via the $\beta$-radius. Connected nodes fall within each other's $\beta$-radius, ensuring exact reconstruction.

expression. A check for full reconstruction is made by comparing $\mathbf{A}$ to $\sigma(\mathbf{X}\mathbf{Y}^\top)$. In practice, $\mathbf{X}\mathbf{Y}^\top$ is clipped to $[0,1]$ (i.e. applying the map $\mathbf{clip} : \mathbb{R} \mapsto [0,1]$ defined as $\mathbf{clip}(x) = \max(0, \min(1, x))$) and the Frobenius norm of the difference between the clipped reconstruction and $\mathbf{A}$ is calculated, $||\mathbf{clip}(\mathbf{X}\mathbf{Y}^\top) - \mathbf{A}||_F / ||\mathbf{A}||_F$. When this measure evaluates to 0, all predicted indices match the actual adjacency matrix and the found embedding thus allows for perfect reconstruction. They carried out this optimization process using a coarse analysis of varying embedding dimensionalities in multiples of 16, and reported the dimensionality of the lowest exact embedding dimension found, e.g., they obtained factorizations of rank 16 on the well-known datasets Cora and Citeseer (Yang et al., 2016), and 32 on ca-HepPh (Leskovec et al., 2005) (see Table 2).

In Gu et al. (2021) the exact embedding dimension was evaluated at the point in which the latent dimension was as large as the number of nodes arguing that this would be the upper bound of required dimensions and the optimal dimensionality defined as the lower dimensional representation concurring within a small margin of such exact embedding. In Bonato et al. (2012); Bonato (2017) the Logarithmic Dimension Hypothesis was proposed arguing for dimensionality of the embedding space of networks scaling as $\mathcal{O}(\log N)$ when considering the MGEO-P model (Bonato et al., 2012) relying on embeddings based on the $L_\infty$-norm in a Blau space in which distance in latent position of nodes are used to characterize their relations (McPherson, 2004) akin to the latent distance model (Hoff et al., 2002). Similarly, in Boratko et al. (2021) it was proven that any directed acyclic graph (DAG) can be perfectly embedded using the probabilistic box embedding in which the probability of observing a link is given by the node-specific box overlap using a $\mathcal{O}(\log N)$ embedding dimension. It was further empirically observed that, for low-dimensional embeddings, metric embedding approaches provided higher capacity embeddings compared to those relying on LPCA (Boratko et al., 2021). Finally, in Chanpuriya et al. (2023) LPCA was generalized to undirected graphs by use of a difference formulation between two symmetric non-negative decompositions bridging the exact network embedding to community detection methodologies. In Chanpuriya et al. (2020) it was proven that for bounded degree (i.e., $k_{\max}$) graphs exact embeddings can be achieved using $D = 2k_{\max} + 1$ dimensions. This result was further refined in Chanpuriya et al. (2023) proving that for sparse networks the arboricity $\alpha$, i.e., largest subgraph density-weighted by the number of nodes in the subgraph, bounds the embedding dimensions by $4\lceil\alpha\rceil^2 + 1$ (Chanpuriya et al., 2023).

## 1.2 CONTRIBUTIONS

From these related works several important open questions remain. Specifically,
**Q1: Can metric model formulations provably provide lower dimensional representations than LPCA?** We presently prove that metric embeddings can uniformly provide at least as low dimensional embeddings as LPCA and highlight empirically that lower dimensional representations of networks exhibiting homophily can be achieved yet the same embedding dimension for networks exhibiting heterophily. **Q2: How can a network's exact low-dimensional embedding efficiently be quantified?** Whereas there exist bounds on the embedding dimensions based on logarithmic scaling wrt. number of nodes (Boratko et al., 2021) in the network, maximum degree (Chanpuriya et al., 2020), and arboricity (Chanpuriya et al., 2023) it is unclear how the lowest possible actual exact embedding dimension of a given network can be identified and Chanpuriya et al. (2020) provided only

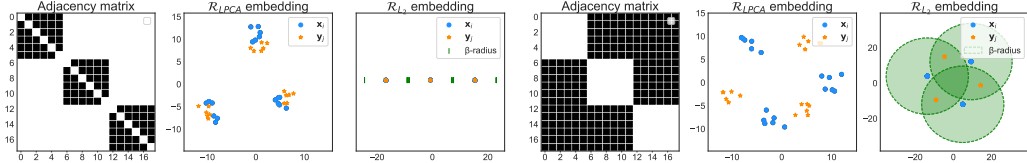

(a) Graph with a homophilous community block structure. Green lines indicate the $L_2$ radius $\beta$.

(b) Graph with a heterophilous block structure. Green circles indicate the $L_2$ radius $\beta$.

Figure 2: Example graphs with a community and an anticommunity structure, respectively, and their corresponding $\mathcal{R}_{LPCA}$- and $\mathcal{R}_{L2}$-embeddings (Lines/Circles denote link thresholds for $\mathcal{R}_{L2}$).

a very coarse assessment of embedding dimensionality. We presently derive an efficient logarithmic search strategy reliably identifying an upper bound on the intrinsic dimensionality of exact network embeddings. **Q3: How can exact network embeddings be linearithmically** $\mathcal{O}(N \log N)$ **quantified in large graphs?** Existing network embedding assessments rely on the explicit evaluation of all links and non-links in the objective function optimized as well as reconstruction check scaling for $N$ nodes as $\mathcal{O}(N^2)$ which is not feasible for large graphs. We presently, explore how metric embeddings enable efficient linearithmic neighborhood queries through the use of KD-trees to scalably assess perfect network reconstruction, and for the first time achieve such a reconstruction for networks with up to a million nodes. We further provide a linearithmic approximation of the full likelihood of metric embeddings providing an efficient learning framework that we demonstrate can be refined using zero-margin hinge-loss optimization over the few approximation-induced missclassified dyads to achieve exact embeddings of large-scale networks.

## 2 METHODS

In the following $\boldsymbol{w}_i$ will denote the $i^{\text{th}}$ row of the matrix $\boldsymbol{W}$ and $\| \cdot \|$ denotes the conventional Euclidean norm. $\text{diag}(\boldsymbol{b})$ will denote a diagonal matrix with the elements of $\boldsymbol{b}$ along the diagonal and $sign(b)$ the conventional sign function that is $-1$ if $b < 0$ and $1$ if $b > 0$. Let $\mathbf{A}$ denote the adjacency matrix of a graph such that $a_{i,j} = 1$ if there is a link between node $i$ and $j$ and $a_{i,j} = 0$ otherwise. Finally let $\tilde{a}_{i,j} = 2a_{i,j} - 1$ be the shifted adjacency matrix such that links and absence of links respectively are given by $1$ and $-1$.

**Q1:** Both the local and global structure of a given graph carry information about the role of each specific node, and this contributes to the complexity of embedding nodes. Modeling the optimization objective such that the resulting embedding space has the capacity to capture these connectivity-related attributes is crucial for encoding the information into low-dimensional node representations. We note two main components in the approach of Chanpuriya et al. (2020), namely a parametrized reconstruction model and a loss measure between the true graph and its reconstruction. We can thus decompose the objective function in Equation 1 into the reconstruction model and loss function respectively given by

$$\mathcal{R}_{LPCA}(\mathbf{X}, \mathbf{Y}) \overset{\Delta}{=} \mathbf{X}\mathbf{Y}^\top, \qquad (2) \qquad \mathcal{L}(\mathbf{R}) \overset{\Delta}{=} \sum_{i,j} -\log \sigma(\tilde{a}_{i,j} r_{i,j}), \qquad (3)$$

where $\tilde{\mathbf{A}} = [\tilde{a}_{i,j}]$ is the shifted adjacency matrix and $\mathbf{R} = [r_{i,j}]$ is the reconstruction of the adjacency matrix, i.e., given by Equation 2. Considering $\mathcal{L}(\cdot)$, in which the underlying modeling assumption is a Bernoulli likelihood, we can express the reconstruction at index $i, j$ as applying thresholding on the link probability, i.e. $p(A_{i,j} = 1) \geq \frac{1}{2}$, or equivalently, $\tilde{a}_{i,j} r_{i,j} \geq 0$.

Analyzing $\mathcal{R}_{LPCA}$, we see that it consists of an outer product between the two embedding matrices. The expressiveness of this reconstruction model thus lies in the pairwise inner products between each row vector in the two matrices. This method is known to have a lot of representational capacity akin to PCA, as the inner product provides a natural similarity measure between vectors, proportional to the angles between them. Notably, the LPCA model formulated above can be considered a generalization to row and column specific embeddings of the latent eigenmodel proposed in Hoff (2007). The eigenmodel also includes a bias term $\beta$, i.e. $\mathcal{R}_{EIG}(\mathbf{X}, \mathbf{Y}, \beta) \overset{\Delta}{=} \beta + \mathbf{X}\mathbf{Y}^\top$. As a result, expressing

this formulation of the latent eigenmodel by LPCA would require an additional dimension, i.e. $[\beta\mathbf{1}\ \mathbf{X}][\mathbf{1}\ \mathbf{Y}]^\top$ provided that $\beta$ is non-zero and $\mathbf{X}\mathbf{Y}^\top$ does not exactly span the constant matrix (a low-probability event). Consequently, the latent eigenmodel formulation can in theory reduce the dimensionality by one when compared to LPCA, i.e. $D^*_{LPCA} - 1 \le D^*_{EIG} \le D^*_{LPCA}$. Similarly, standard PCA requires an extra dimension if the data is not centered, i.e., when applying SVD directly to an uncentered data matrix. Importantly, we are only arguing for the dimensionality of the solution and not that the solution must be exactly $[\beta\mathbf{1}\ \mathbf{X}][\mathbf{1}\ \mathbf{Y}]^\top$ since the model reconstructions $\mathcal{R}_{LPCA}, \mathcal{R}_{EIG}$ , and $\mathcal{R}_{L_2}$ for LPCA, LEIG, and LDM are non-unique and can all be modified by an orthogonal matrix $\mathbf{Q}$ such that $\tilde{X} = X\mathbf{Q}$ and $\tilde{Y} = Y\mathbf{Q}^{-1}$ provide identical reconstructions.

To further improve the expressiveness of the node embedding optimization objective, we consider the Euclidean latent distance model (LDM) (Hoff et al., 2002; Hoff, 2007) that provides a metric specification of the reconstruction model according to

$$\mathcal{R}_{L_2}(\mathbf{X}, \mathbf{Y}, \beta) \triangleq \left[\beta - \|\boldsymbol{x}_i - \boldsymbol{y}_j\|_2\right]. \tag{4}$$

The LDM is a metric model, i.e., it yields an embedding vector space endowed with a distance function, which has useful properties we will elaborate on later. Importantly, in Theorem 2.1, we establish that the $\mathcal{R}_{L_2}$-reconstruction model can provide more favorable embedding dimensions using up to two embedding dimensions less than $\mathcal{R}_{LPCA}$. A model overview is given in Figure 1.

**Theorem 2.1.** *Let $D^*_{LPCA}$ and $D^*_{L_2}$ denote the lowest exact embedding dimension for a graph embedding obtainable by optimization w.r.t. the $\mathcal{R}_{LPCA}$-reconstruction and $\mathcal{R}_{L_2}$-reconstruction respectively. We then have the relationship*

$$D^*_{LPCA} - 2 \le D^*_{L_2} \le D^*_{LPCA}. \tag{5}$$

For a proof of the above theorem see section A.2 in the supplementary material.

Importantly, metric models are especially efficient when modeling homophily in networks, i.e., a friend of a friend is also a friend, which is naturally entailed by the triangular inequality using metric embeddings (Hoff et al., 2002). On the other hand, heterophilous networks (Chanpuriya et al., 2023) also defined in terms of stochastic equivalence (Hoff, 2007) in which dissimilar nodes can be grouped is well accounted for by LPCA and can according to the above theorem at least with similar embedding dimensions be accounted for by the LDM. This is illustrated in Figure 2 where we compare the $\mathcal{R}_{LPCA}$ and $\mathcal{R}_{L_2}$ considering a homophilous community structured and heterophilous block-structured network. We here observe that the $\mathcal{R}_{L_2}$ formulation can account for communities using an embedding dimension of $D = 1$ whereas $\mathcal{R}_{LPCA}$ and $\mathcal{R}_{LEIG}$ (not shown) requires $D = 2$ whereas all approaches can perfectly reconstruct the heterophilous network using $D = 2$. In Figure 3 we further highlight an example empirically verifying the bounds of Theorem 2.1. $\mathcal{R}_{LPCA}$ requires three dimensions since more than three communities (as in Figure 2) need angles greater than 90 degrees in 2D for perfect reconstruction, which is not possible for the 10 communities in Figure 3. As a result, an additional dimension is needed to mimic the bias term included in $\mathcal{R}_{LEIG}$ that can reconstruct the network using two dimensions exploiting that the bias term can threshold on angles less than 90 degrees for perfect reconstruction. Importantly, $\mathcal{R}_{L_2}$ can trivially embed this network using only one dimension due to its metric properties.

**Q2:** To find an upper bound for the low-dimensional exact embedding, we propose Algorithm 1, which utilizes binary search (also called logarithmic search due to its logarithmic scaling) to search through an interval $[lb, ub], lb < ub \in \mathbb{Z}^+$ of embedding space ranks with linearithmic time complexity (Knuth, 1998). We initialize the first iteration of embeddings at a relatively high dimension $D_0 = ub$, yielding the embedding matrices $\mathbf{X}, \mathbf{Y} \in \mathbb{R}^{N \times D_0}$. When the LDM optimization yields embedding matrices that perfectly reconstruct the graph, they are concatenated as $\mathbf{Z} = [\mathbf{X}\ \ \mathbf{Y}]^\top \in \mathbb{R}^{2N \times D_0}$. As the LDM is translation invariant, we center the concatenated matrix as $\mathbf{Z}_c = \mathbf{Z} - \mathbf{1}\boldsymbol{\mu}_Z^\top$, where $\boldsymbol{\mu}_Z$ is a vector containing the rowwise mean of $\mathbf{Z}$. We then proceed to the next iteration of embeddings $\mathbf{X}', \mathbf{Y}' \in \mathbb{R}^{N \times D}$, where the new rank $D$ is chosen according to binary search. We initialize the new embeddings using the low-rank SVD with $D$ dimensions, given as $\mathbf{Z}_c \approx \mathbf{U}_D\mathbf{\Sigma}_D\mathbf{V}_D^\top$, to project the previous embedding matrices onto the reduced space as $\mathbf{X}' = (\mathbf{X} - \mathbf{1}\boldsymbol{\mu}_Z^\top)\mathbf{V}_D, \mathbf{Y}' = (\mathbf{Y} - \mathbf{1}\boldsymbol{\mu}_Z^\top)\mathbf{V}_D \in \mathbb{R}^{N \times D}$. This is repeated by updating $D$ in accordance with the binary search. In Figure 7 of the supplementary material the benefits of truncated SVD initialization from a higher embedding dimension as opposed to randomly initializing the new embedding at the given embedding dimension are illustrated.

**Q3:** A metric space is defined as a set $M$ endowed with a distance function $d : M \times M \to \mathbb{R}$ satisfying the metric properties for points $x, y, z \in M$, i.e. **(I)** $d(x,x) = 0$, **(II)** for $x \neq y : d(x,y) > 0$, **(III)** $d(x,y) = d(y,x)$ and **(IV)** [the triangle inequality]: $d(x,z) \leq d(x,y) + d(y,z)$. Since the $\mathcal{R}_{L_2}$-reconstruction is based on the Euclidean distance between the latent coordinates of the nodes, the set of nodes jointly embedded during training will therefore satisfy these metric properties.

Optimizing the objective function with the $\mathcal{R}_{L_2}$-reconstruction as in Equation 4, we obtain a set of parameters $\boldsymbol{\theta}^\star = \{\mathbf{X}^\star, \mathbf{Y}^\star, \beta^\star\}$. Assuming $\boldsymbol{\theta}^\star$ corresponds to a perfect reconstruction and noting that $\sigma(\tilde{a}_{i,j} [\mathcal{R}_{L_2}(\boldsymbol{\theta}^\star)]_{i,j}) > 0$, we observe that $\text{sign}([\mathcal{R}_{L_2}(\boldsymbol{\theta}^\star)]_{i,j})$ is $+1$

---

**Algorithm 1** Progressive search for solution with lowest EED

**Require:** $lb, ub \in \mathbb{Z}^+$ and $lb < ub$.
1: Initialize search interval as $[lb, ub]$
2: $D^\star \leftarrow$ None         ▷ Optimal $D^\star$ found.
3: $D_0 \leftarrow ub$         ▷ Initial candidate EED.
4: $\boldsymbol{\theta}_0 \leftarrow \mathbb{R}^{N \times D_0} \times \mathbb{R}^{N \times D_0}, \theta_{i,j} \sim \mathcal{N}(0,1)$   ▷ For $\mathcal{L}_2, \beta \leftarrow \beta_0 \sim \text{Unif}(0,1)$
5: Initialize $M_\mathcal{R}$ with $\boldsymbol{\theta}_0$    ▷ $M_\mathcal{R}$: model w. reconstruction $\mathcal{R}$.
6: $\boldsymbol{\theta} \leftarrow$ Train $M_\mathcal{R}$ until convergence or full reconstruction, otherwise stop search.
7: **while** $lb \leq ub$ **do**
8:     $D \leftarrow \left\lfloor \frac{lb+ub}{2} \right\rfloor$
9:     $[U, \Sigma, V] \leftarrow \text{SVD}(\text{Concat}[X,Y])$ ▷ For $\mathcal{L}_2$, center embedding space after concat.
10:     $(X', Y') \leftarrow (V_{:D} X, V_{:D} Y)$
11:     Initialize $M'_\mathcal{R}$ with $(X', Y')$
12:     $\boldsymbol{\theta}' \leftarrow$ Train $M'_\mathcal{R}$ until convergence or exact embedding
13:     **if** exact embedding achieved **then**
14:        $D^\star \leftarrow D$
15:        $ub \leftarrow D - 1$
16:        $\boldsymbol{\theta} \leftarrow \boldsymbol{\theta}'$
17:        **Stop** search if $D = lb$.
18:     **else**
19:        **Stop** search if $D = ub$.
20:        $lb \leftarrow D + 1$
21:     **end if**
22: **end while**

---

for links and $-1$ for nonlinks, and thus $\beta$ can be viewed as the radius of the hypersphere, centered in $\boldsymbol{x}_i^\star$, which encapsulates all $\boldsymbol{y}_j^\star$ for which the node index pair $i, j$ constitute a link in the embedded graph. This corresponds to a unit disk graph (Clark et al., 1990) defined across two embeddings $\mathbf{X}, \mathbf{Y}$ instead of a single embedding $\mathbf{X}$. Examining the full reconstruction $\mathbf{R} = \mathcal{R}_{L_2}(\mathbf{X}, \mathbf{Y}, \beta)$, we see that the $\boldsymbol{x}_i$ and $\boldsymbol{y}_j$ node embeddings encode the source and target nodes, respectively, in the pairwise relations between nodes in the graph.

**Checking for perfect reconstruction:** To check if an exact embedding has been found, we have to check if the reconstructed adjacency matrix $\hat{\mathbf{A}}$ from the embeddings is the same as the original $\mathbf{A}$. This is done by calculating the Frobenius error between them $||\hat{\mathbf{A}}-\mathbf{A}||_F/||\mathbf{A}||_F$ which poses issues with both runtime and memory usage when working with large graphs as it requires the dense adjacency matrix. For this reason, we propose a different approach to check if an exact embedding has been found exploring that the metric properties allow us to employ different similarity searching techniques (Zezula et al., 2006; Yianilos, 1993). In particular, we use the fixed-radius nearest neighbors search based on KD-trees which can be constructed in linearithmic time (Knuth, 1998; Friedman et al., 1977; Bentley, 1975). If we find all the nearest neighbors within the distance $\beta$ in the embedding space, we can compare the found neighbors with the sparsely represented edge index lists. The worst case runtime for making the comparison would be $\mathcal{O}(N^2)$, which happens in the case that everything in the graph is connected. As large graphs are mostly very sparse (Barabási & Pósfai, 2016) with edges typically scaling sub-linearithmically the effective runtime will be much lower (Nakis et al., 2022).

**Scalable inference:** When working with very large graphs a lot of memory will be required to store the full dense adjacency matrices, which is necessary for the reconstruction check using Frobenius error. The reconstruction check can be done with only the sparse representation of the proposed nearest neighbors full reconstruction check, but we still need to address the issue of memory when calculating the loss and learning the representation. Two sampling methods that try to get around the issue of memory on very large graphs are random node sampling (Leskovec & Faloutsos, 2006) and negative sampling/case-control inference (Hamilton, 2020; Raftery et al., 2012).

**Random Node (RN) sampling:** In random node (RN) sampling (Leskovec & Faloutsos, 2006) a random set of nodes is sampled uniformly amongst the original $N$ nodes. We let $b$ denote the set of sampled node indices, the optimization objective in Equation 3 can be reformulated as: $\mathcal{L}_{\text{RN}}(R) \overset{\Delta}{=} -\sum_{i \in b} \sum_{j \in b} \log \sigma(\tilde{a}_{i,j} r_{i,j})$. This is equivalent to inducing a subgraph from the nodes in $b$ and performing an optimization step on this subgraph as if it was the original adjacency matrix. When using RN sampling where each node has an equal probability of being in the induced sample, we might end up not preserving the structural properties of the graph. In Stumpf et al. (2005) they show that RN sampling does not retain the power law degree distribution for scale-free networks.

**Case-control (CC) sampling:** Case control (Raftery et al., 2012) or negative sampling as formulated in Mikolov et al. (2013); Goldberg et al. (2014) and also explained in Hamilton (2020) considers all nodes in the graph which we denote $\mathcal{V}$. For each of the node indices $i'$ all the links are collected $l_1^{(i')}$ and for $k \in \mathbb{Z}^+$ sample $k \cdot \left| l_1^{(i')} \right|$ non-links uniformly, denoted by $l_0^{(i')}$. In practice, we set $k = 5$.

Defining $l_1 = \bigcup_{i' \in \mathcal{V}} \{ l_1^{(i')} \}$ and $l_0 = \bigcup_{i' \in \mathcal{V}} \{ l_0^{(i')} \}$ the loss from Equation 3 can be redefined as:

$$\mathcal{L}_{\text{CC}}(R) \triangleq \sum_{r_i \in \hat{l}_1} (-\log \sigma(r_i)) + \sum_{r_j \in \hat{l}_0} (-w_j \log \sigma(r_j)) \tag{6}$$

Where $\hat{l}_1$ and $\hat{l}_0$ corresponds to the sets of links and non-links in $l_1$ and $l_0$, e.g. $\mathcal{R}_{L_2}$. $w_j = \frac{N - |l_1^{(j)}|}{|l_0^j|}$ is a node specific recalibration weight based on the amount of links and non-links.

**Hierarchical Block Distance Model approximation and hinge loss active set optimization:** Sampling methods visit only a very small fraction of the network during training which can especially become an issue for exact reconstruction of large-scale networks potentially creating problems with convergence of the model, as well as convergence speed. We therefore propose a two-stage optimization approach based on the linearithmically scaling ($\mathcal{O}(N \log(N))$) hierarchical block distance model (HBDM) originally proposed in the context of the Poisson likelihood (Nakis et al., 2022). To achieve perfect reconstruction from an HBDM initialized solution we further exploit that the hinge loss reduces the loss function to only consider missclassified dyads but with same stationary points for exact network reconstruction as Equation 3. Specifically, we start by optimizing the HBDM based on the following augmentation of the method to the Bernoulli log-likelihood:

$$\mathcal{L}_{\text{HBDM}}(R) \triangleq \sum_{\substack{i \neq j \\ y_{i,j} = 1}} \left( \beta - ||\mathbf{x}_i - \mathbf{y}_j||_2 \right) - \sum_{k_L = 1}^{K_L} \left( \sum_{i,j \in C_{k_L}} \log(1 + \exp(\beta - ||\mathbf{x}_i - \mathbf{y}_j||_2)) \right)$$
$$- \sum_{l=1}^{L} \sum_{k=1}^{K_l} \sum_{k' \neq k}^{K_l} \left( \log(1 + \exp(\beta - ||\boldsymbol{\mu}_k^{(l)} - \boldsymbol{\mu}_{k'}^{(l)}||_2)) \right), \tag{7}$$

in which a hierarchical structure akin to a KD-tree is used to reduce the softplus contribution from the likelihood based on approximating this part of the likelihood using an optimally learned hierarchical structure based on the current learned embedding space thereby providing an accurate approximation of the full likelihood (Nakis et al., 2022). By optimizing the embedding space using HBDM only a relatively small number of dyads will remain misclassified as a result of the block approximation. The Hierarchical Block Distance Model (HBDM) uniquely characterizes the entire likelihood of large-scale graphs without sampling, achieving linearithmic space and time complexity through a hierarchical approximation of the total likelihood via metric clustering under Euclidean distance. The sums over terms $l, k, L, K$ refer to the $K$ clusters and $L$ layers used by HBDM, while $Y_{N \times N} = (y_{i,j}) \in {0, 1}^{N \times N}$ represents the adjacency matrix of the graph, where $y_{i,j} = 1$ if the pair $(i, j) \in E$, otherwise it is 0 for all $1 \leq i, j \leq N$. Our model uses HBDM for initializing its latent space to provide a "hot" start.

In the second stage, we explore that for perfect reconstruction the stationary points of the Bernoulli likelihood and hinge loss are the same. This enables to use a zero-margin hinge loss optimization procedure that operates only over the active set of misclassified dyads (identified using the linearithmic reconstruction check) as opposed to all dyads in the logistic loss Equation 3 to efficiently achieve perfect reconstruction:

$$\mathcal{L}_{\text{HL}}(R) \triangleq \sum_{(i,j) \in \mathcal{S}} \max(0, -\tilde{a}_{i,j} r_{i,j}), \tag{8}$$

where $\mathcal{S}$ is the active set. The active set $\mathcal{S}$ is then updated after each iteration via the KD-tree-based perfect reconstruction check until perfect reconstruction is achieved. This HBDM initialization allows our model to misclassify very few pairs, denoted as $M$, with $M \ll N^2$ (observed empirically) for large graphs. The model then iterates only over these misclassified pairs, identified using KD-trees and the metric properties of the LDM. These pairs are easily accounted for by the proposed analytical hinge loss optimization without computational constraints. Consequently, our model can achieve perfect reconstruction of large-scale graphs efficiently. To our knowledge, we are the first to achieve

Table 1: Graphs used in the experiments along with some statistics. Comp. Arboricity denotes the maximal arboricity obtained considering as subgraphs only the connected components of the graph as the actual arboricity requires infeasible exhaustive evaluation of all combinations of subsets of nodes thus providing a lower bound on the arboricity.

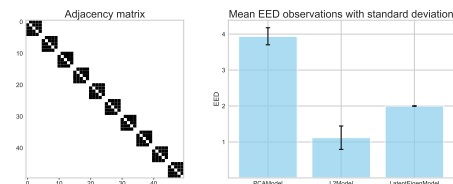

Figure 3: Visualization of the training statistics over 100 test runs on the synthetic block graph seen in the left figure. The bar is the mean exact embedding dimension (EED) and the error bars correspond to the standard deviation of the measurements. An extended version of this figure can be seen in the supplementary A.6.

| Dataset | Nodes | Type | Avg. Degree | Max Degree / Comp. Arboricity | Connected Components | Total Triangles |
|---|---|---|---|---|---|---|
| Cora | 2708 | undirected | 3.90 | 168 / 5.75 | 78 | 1630 |
| CiteSeer | 3327 | undirected | 2.74 | 99 / 4.0 | 438 | 1167 |
| Facebook | 4039 | directed | 21.85 | 1043 | 1 | 1612010 |
| ca-GrQc | 5242 | undirected | 5.53 | 81 / 7.0 | 355 | 48260 |
| wiki-Vote | 7115 | directed | 14.57 | 893 | 24 | 608389 |
| p2p-Gnutella04 | 10876 | directed | 3.68 | 100 | 1 | 934 |
| ca-HepPh | 12008 | undirected | 19.74 | 491 / 21.0 | 278 | 3358499 |
| PubMed | 19717 | undirected | 4.50 | 171 / 4.5 | 1 | 12520 |
| com-amazon | 334863 | undirected | 5.53 | 549 / 5.53 | 1 | 667129 |
| roadNet-PA | 1088092 | undirected | 2.83 | 9 / 4.0 | 206 | 67150 |

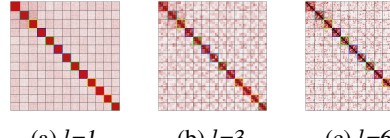

(a) $l=1$     (b) $l=3$     (c) $l=6$

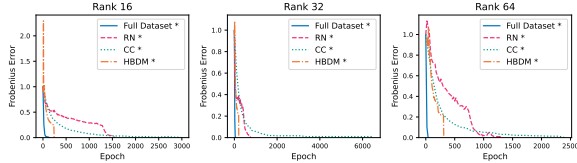

Figure 4: Reordered adjacency matrices of the *com-amazon* network at different hierarchy levels learned by HBDM.

Figure 5: Comparison of case-control (CC), random node sampling (RN), HBDM, and full likelihood inference on the Cora network.

this and provide such an efficient optimization procedure. The HBDM has linearithmic complexity, and we have empirically observed that after very few HBDM iterations, the number of misclassified pairs is in the linearithmic scale. This results in an overall linearithmic complexity for our proposed method, making it a highly scalable method, much more efficient than competing baselines.

## 3 RESULTS AND DISCUSSION

The methods introduced are tested on commonly used graph datasets: Cora, Citeseer, Pubmed (Yang et al., 2016), ca-HepPh (Leskovec et al., 2005; Gehrke et al., 2003), and ca-GrQc (Leskovec et al., 2007) which are all different publication and citation networks, Facebook (Yang et al., 2020) which is a subset of the Facebook network, wiki-Vote (Leskovec et al., 2010b;a) consisting of voting on Wikipedia, the peer to peer network p2p-Gnutella04 (Leskovec et al., 2007; Ripeanu et al., 2002), the Amazon network (Yang & Leskovec, 2012), and Roadnet-PA which is a road network of Pennsylvania (Leskovec et al., 2008). Many of the graphs are retrieved from the SNAP Datasets collection (Leskovec & Krevl, 2014) or PyTorch Geometric (Fey & Lenssen, 2019). The datasets are all preprocessed by ignoring edge weights, i.e., each entry in the adjacency graph is either 0 or 1. We list graph statistics for the datasets in Table 1, as well as note the edge type, i.e., directed/undirected. Optimization is performed using the ADAM optimizer (Kingma & Ba, 2014), with the learning rate halved after $k$ steps of no improvement. The initial learning rate and $k$ vary by loss function and dataset, assessed qualitatively. Experiments run for up to 30,000 epochs, with rank intervals adjusted per dataset and prior results (Chanpuriya et al., 2020). See the supplementary material for details.

**Searching for the optimal embedding dimension** $D^*$: 5 searches have been carried out for each of the datasets and the minimum exact embedding dimension ($D^*$) has been reported along with the mean and standard deviation across the analyses of each network. These results are presented in Table 2 along with the exact embedding dimensions reported by Chanpuriya et al. (2020). Notably, in Chanpuriya et al. (2020) all networks were analyzed as undirected whereas we presently preserve the directed structure of the directed networks to highlight that the directionality of links can naturally be accounted for by the exact embedding. As a result, the results can only be directly compared for the five undirected networks. In Chanpuriya et al. (2020) self-links were further included in the modeling, but we presently do not model self-links ignoring these in the loss function. However,

Table 2: Lowest exact embedding dimensions ($D^*$) found for each dataset across 5 searches along with the mean and standard deviations across the searches. We have marked directed networks with a "*" as these will not be comparable with Chanpuriya et al. (2020) as they converted all networks to undirected networks.

Table 3: Lowest exact embedding dimensions ($D^*$) found for the two large-scale networks, com-amazon and roadNet-PA.

| Dataset | $D^*$ ($L_2$) | | $D^*$ (LPCA) | | $D^*$ (Eigenmodel) | | $D^*$ ($L_2$, hinge loss) Margin of 0 | | $D^*$ Chanpuriya et al. |
|---|---|---|---|---|---|---|---|---|---|
| Cora | **6** | (6.2 $\sigma$ 0.45) | 9 | (9.8 $\sigma$ 0.45) | 9 | (9.4 $\sigma$ 0.85) | **7** | (7 $\sigma$ 0) | 16 |
| Citeseer | **6** | (6.7 $\sigma$ 0.55) | 9 | (9.2 $\sigma$ 0.45) | 9 | (9.2 $\sigma$ 0.45) | **7** | (7 $\sigma$ 0) | 16 |
| Facebook* | **20** | (20.67 $\sigma$ 0.52) | 22 | (22.8 $\sigma$ 0.45) | 21 | (22.6 $\sigma$ 0.89) | **20** | (20 $\sigma$ 0) | - |
| ca-GrQc | **8** | (8 $\sigma$ 0) | 10 | (10.8 $\sigma$ 0.45) | 10 | (10.4 $\sigma$ 0.55) | **8** | (8 $\sigma$ 0) | 16 |
| Wiki-Vote* | **41** | (42.33 $\sigma$ 1.97) | 45 | (45.8 $\sigma$ 0.45) | 45 | (46.4 $\sigma$ 1.34) | 42 | (42.2 $\sigma$ 0.45) | 48 |
| p2p-Gnutella04* | **14** | (14 $\sigma$ 0) | 17 | (17.8 $\sigma$ 0.45) | 17 | (18 $\sigma$ 0.71) | 16 | (16 $\sigma$ 0) | 32 |
| ca-HepPh | **16** | (16 $\sigma$ 0) | 19 | (19.8 $\sigma$ 0.84) | 19 | (19.4 $\sigma$ 0.55) | **16** | (16.67 $\sigma$ 0.52) | 32 |
| Pubmed | **14** | (14 $\sigma$ 0) | 17 | (17.8 $\sigma$ 0.45) | 17 | (17.4 $\sigma$ 0.55) | 16 | (16 $\sigma$ 0) | 48 |

| | com-amazon | roadNet-PA |
|---|---|---|
| $D^*$ ($L_2$) | 13 | 16 |

in the supplementary A.3, we include the modeling of self-links for direct comparison and find that this has little influence on the extracted embedding dimensionality $D^*$. From the results we see that using the proposed efficient search scheme, we identify the existence of solutions with substantially lower optimal embedding dimension $D^*$ than previously reported and substantially lower than the bounds in terms of maximal degree $D = 2k_{\max} + 1$ (Chanpuriya et al., 2020) and arboricity $4\lceil\alpha\rceil^2 + 1$ (Chanpuriya et al., 2023) as presently evaluated only across each component of the graphs. Additionally, these results align with the theoretical result from Theorem 2.1, i.e., $D^*$ for the $\mathcal{R}_{L_2}$-embedding is at least as good as the $\mathcal{R}_{LPCA}$-embedding. For Cora, the $\mathcal{R}_{LPCA}$ solution has a $D^*$ that is three dimensions larger than the $\mathcal{R}_{L_2}$-embedding. This is likely due to the optimization being increasingly difficult for lower dimensions, causing the training process to get stuck in local minima rather than finding the expected (or approximate) optimal solutions, as we according to Theorem 2.1 can analytically extract an embedding of dimension $D^*_{LPCA} = D^*_{L2} + 2$ directly from the $\mathcal{R}_{L_2}$ solution.

**Instability of training procedure:** As with many other machine learning algorithms, optimizing for the embeddings is dependent on initialization and hyperparameter settings, i.e., number of epochs, learning rate, etc. As a result there are no guarantees that the lowest possible $D^*$ is identified. The optimization procedure is initialized with random embedding matrices, and this can lead to the model getting stuck in local minima. We showcase this behavior using synthetic data in Figure 3. We run the training procedure for each of the $\mathcal{R}_{L_2}$-, $\mathcal{R}_{EIG}$- and $\mathcal{R}_{LPCA}$-models, where we start by testing if we can embed it in 1 dimension, moving up to 2 dimensions if not possible, and so on, until convergence. We do this 100 times for each model. We observe that $\mathcal{R}_{L_2}$, and $\mathcal{R}_{EIG}$ reliably extract their respective lowest embedding dimensions of $D = 1$ and $D = 2$ whereas the $\mathcal{R}_{LPCA}$-model is less reliable and only in a few of the runs correctly identifies the $D = 3$ solution.

**Statistics of the reconstructed graph for embedding dimensions below the optimal $D^*$:** In Figure 6, we present graph statistics for the *Cora* network, to illustrate the impact of additional compression on the reconstructed graph as $D < D^*$. Notably, the statistics for $D^*$ represent the ground truth, reflecting perfect graph reconstruction. Figure 6a illustrates how the average degree of the reconstructed network evolves as the network is compressed into lower-dimensional embedding spaces than the lowest exact dimension. Figure 6b demonstrates that the average shortest path length of the reconstructed graph decreases with increasing compression, reflecting the network's structural simplification. The graph density, as provided in Figure 6c, is consistent with the observed increase in average degree, also rises as the network becomes more compressed. Finally, Figure 6d highlights the increasing percentage of misclassified dyads as the model is further compressed, emphasizing its diminishing ability to accurately reconstruct the original network. (See also supplementary A.10.)

**Scalable exact network embeddings:** In Figure 5, we compare case-control (CC) inference, random node (RN) sampling, and the HBDM inference to inference using the full likelihood considering the Cora network. We observe that the HBDM outperforms the alternative similar linearithmic $\mathcal{O}(N \log N)$ specified CC and RN sampling procedures providing scalable inference while achieving convergence relatively close to the convergence using the full likelihood. In Table 3 the achieved exact embedding dimension of the two large networks are given. This demonstrates that the two-phase procedure is scalable, but due to computational costs, we leave the search for a tighter bound on their embedding dimension for future work. In Figure 4 we visualize the Amazon network organized according to the exact embedding using the corresponding HBDM induced hierarchical approximation structure of this embedding. The exact embedding effectively captures the underlying

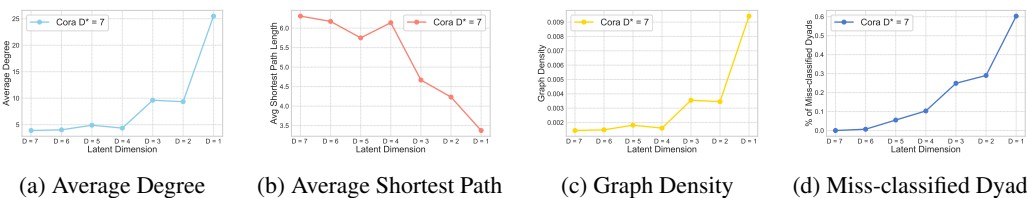

(a) Average Degree     (b) Average Shortest Path     (c) Graph Density     (d) Miss-classified Dyads

Figure 6: Graph statistics for the reconstructed graph as the latent dimension decreases from the exact embedding dimension $(D^*)$ to $(D = 1)$ for *Cora*. $(D^*)$ ensures perfect reconstruction.

community structure with high within-block densities, while progressively revealing more detailed network structures as we traverse the hierarchy.

## 4   CONCLUSION, LIMITATIONS AND BROADER IMPACT

Our work explores the intrinsic dimensionality of graphs by posing the question of just how few dimensions are needed to represent a graph in a way that allows for perfect reconstruction. Through metric models, we demonstrate both theoretically and empirically that we can construct more expressive embedding spaces than LPCA (Chanpuriya et al., 2020) and at the same time exploit metric properties to linearithmically, $\mathcal{O}(N \log N)$, scale the inference of exact network embeddings to large graphs. Using our framework we achieve substantially lower embedding dimensions than previously reported. Surprisingly, we also observe from our large-scale network analysis that even large networks can be modeled exactly using very low-dimensional representations. We anticipate our efficient exact network embeddings can have wide applications within network science from the visualization and extraction of communities as presently highlighted to graph representation learning tasks including node classification and link prediction in which the LDM based on Euclidean distance has demonstrated strong performance using low-dimensional representations (Nakis et al., 2022). Furthermore, we anticipate that exact embeddings can have important applications within effcient network motif discovery (Milo et al., 2002; Vespignani, 2003) to the characterization of network resilience and path properties based on the extracted topology induced by the exact network embedding. The approach can also be used to identify the interpolation threshold which has previously been used to define comparatively suitable network representations (Gu et al., 2021).

Our approach does not necessarily identify the lowest possible network embedding dimension but an upper bound $D^*$ and can be prone to issues of local minima, see also section 3. We presently considered metric embedding by the $L_2$-norm as originally proposed for the latent distance model (Hoff et al., 2002) which enabled us to establish a direct relationship to the embedding dimensions of LPCA. However, we note that other metric embedding approaches for instance as proposed relying on the $L_\infty$ norm (Bonato et al., 2012; Bonato, 2017; Boratko et al., 2021) or hyperbolic geometry (Krioukov et al., 2009; Boguñá et al., 2010; Papadopoulos et al., 2012; Thomas et al., 2016; Muscoloni et al., 2017; Nickel & Kiela, 2017b; 2018) may enable lower dimensional representations than the ones presently achieved. In the supplementary material A.1, we contrast the performance of the Euclidean embeddings to embeddings based on the Poincaré disk model, finding that the two embedding approaches perform very similar in terms of the estimated exact embedding dimensionality. Notably, our scalable inference procedure directly generalizes to other metric embedding approaches and future work should further investigate properties of other choices of geometry on the extracted dimensionality $D^*$.

Whereas we expect exact embeddings to be useful for a variety of graph representation learning tasks care has to be taken in the context of link prediction as exact embeddings will imply learning explicitly to also characterize links set to zero as zero. As such, the approach may only provide meaningful predictions when treating links as missing from the loss function (using the hold-out method) if used for link-prediction (Miller et al., 2009) as opposed to the traditional approach setting links to zero and predicting that they were changed (Liben-Nowell & Kleinberg, 2003). Low-dimensional exact embeddings can be directly used for network visualization and low-dimensional representation as presently highlighted. This however can also be used for surveillance purposes in which the compact representations induced by the embeddings provide low-dimensional node-specific fingerprints which can be used for profiling and identification purposes.

ETHICS STATEMENT

The authors declare no conflicts of interest.

REPRODUCIBILITY STATEMENT

All code for reproducing the experiments is available at: https://github.com/AndreasLF/HowLowCanYouGo. Furthermore, all data sets used in the experimentation are publicly available.

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

# A   APPENDIX

## A.1   EXPERIMENTS WITH A HYPERBOLIC DISTANCE MODEL

To verify that our method extends to other metric spaces than the Euclidean, we run the EED search procedure using a hyperbolic distance model, based on the Poincaré ball model as seen in ((Nickel & Kiela, 2017a)), as this model formulation is well-suited for gradient-based optimization. Concretely, we optimize using the Riemannian ADAM implementation from the `geoopt` library(Kochurov et al., 2020), which is based on (Becigneul & Ganea, 2019).

Table 4: Extended version of Figure 2. Lowest exact embedding dimensions ($D^*$) found for each dataset across 5 searches along with the mean and standard deviations across the searches, including the Hyperbolic distance model. See Figure 1 for graph statistics for each of the graphs. Results using hinge loss are obtained with a margin of 0. **Currently only 3 searches have been performed for the Hyperbolic model on Pubmed.

| Dataset | $D^*$ ($L_2$) | | $D^*$ (LPCA) | | $D^*$ (Eigenmodel) | | $D^*$ ($L_2$) Hinge loss | | $D^*$ Chanpuriya et al. | $D^*$ (Hyperbolic) Hinge loss | |
|---|---|---|---|---|---|---|---|---|---|---|---|
| Cora | **6** | (6.2 $\sigma$ 0.45) | 9 | (9.8 $\sigma$ 0.45) | 9 | (9.4 $\sigma$ 0.85) | 7 | (7 $\sigma$ 0) | 16 | 7 | (7.4 $\sigma$ 0.49) |
| Citeseer | **6** | (6.7 $\sigma$ 0.55) | 9 | (9.2 $\sigma$ 0.45) | 9 | (9.2 $\sigma$ 0.45) | 7 | (7 $\sigma$ 0) | 16 | 7 | (7.8 $\sigma$ 0.40) |
| Facebook* | **20** | (20.67 $\sigma$ 0.52) | 22 | (22.8 $\sigma$ 0.45) | 21 | (22.6 $\sigma$ 0.89) | **20** | (20 $\sigma$ 0) | - | **20** | (20.6 $\sigma$ 0.49) |
| ca-GrQc | **8** | (8 $\sigma$ 0) | 10 | (10.8 $\sigma$ 0.45) | 10 | (10.4 $\sigma$ 0.55) | **8** | (8 $\sigma$ 0) | 16 | **8** | (8.8 $\sigma$ 0.40) |
| Wiki-Vote* | **41** | (42.33 $\sigma$ 1.97) | 45 | (45.8 $\sigma$ 0.45) | 45 | (46.4 $\sigma$ 1.34) | 42 | (42.2 $\sigma$ 0.45) | 48 | 45 | (46.0 $\sigma$ 1.55) |
| p2p-Gnutella04* | **14** | (14 $\sigma$ 0) | 17 | (17.8 $\sigma$ 0.45) | 17 | (18 $\sigma$ 0.71) | 16 | (16 $\sigma$ 0) | 32 | 17 | (17.6 $\sigma$ 0.49) |
| ca-HepPh | **16** | (16 $\sigma$ 0) | 19 | (19.8 $\sigma$ 0.84) | 19 | (19.4 $\sigma$ 0.55) | **16** | (16.67 $\sigma$ 0.52) | 32 | 18 | (18.6 $\sigma$ 0.80) |
| Pubmed | **14** | (14 $\sigma$ 0) | 17 | (17.8 $\sigma$ 0.45) | 17 | (17.4 $\sigma$ 0.55) | 16 | (16 $\sigma$ 0) | 48 | 18 | (18.67 $\sigma$ 0.47)** |

## A.2   PROOF OF THEOREM 2.1

In this section we provide the proof of Theorem 2.1.

**Theorem A.1.** *Let $D^*_{LPCA}$ and $D^*_{L_2}$ denote the lowest possible exact embedding dimension for a graph embedding obtained by optimization w.r.t. the $\mathcal{R}_{LPCA}$-reconstruction and $\mathcal{R}_{L_2}$-reconstruction respectively. Then:*

$$D^*_{LPCA} - 2 \leq D^*_{L_2} \leq D^*_{LPCA} \tag{9}$$

*Proof.* Recall that the probability of a link at index pair $(i, j)$ in LPCA is given by $p(A_{i,j}) = \sigma\left(\left[\mathbf{X}\mathbf{Y}^\top\right]_{i,j}\right)$ and recall that this implies that a link exists if and only if $\left[\mathbf{X}\mathbf{Y}^\top\right]_{i,j} > 0$.

We note that the signs of $\mathbf{X}$ and $\mathbf{Y}$ are invariant to scaling by positive scalars, and we define $\boldsymbol{\gamma} := [\gamma]_i$ and $\boldsymbol{\alpha} := [\alpha]_j$, where $\gamma_i, \alpha_j \in \mathbb{R}^+$. We can now write:

$$\text{sign}\left(\mathbf{X}\mathbf{Y}^\top\right) = \text{sign}\left(\text{diag}\left(\boldsymbol{\gamma}\right)\mathbf{X}\mathbf{Y}^\top\text{diag}\left(\boldsymbol{\alpha}\right)\right) = \text{sign}\left(\tilde{\mathbf{X}}\tilde{\mathbf{Y}}^\top\right).$$

From this expression, we can ensure row-wise normalization of the factorization matrices, i.e. $\left\|\tilde{\boldsymbol{x}}_i^{(\text{row})}\right\|_2 = \left\|\tilde{\boldsymbol{y}}_j^{(\text{row})}\right\|_2 = 1$, by setting $\gamma_i = \left\|\boldsymbol{x}_i^{(\text{row})}\right\|_2^{-1}$ and $\alpha_j = \left\|\boldsymbol{y}_j^{(\text{row})}\right\|_2^{-1}$.

Considering the metric embeddings, the distance $\|\tilde{\boldsymbol{x}}_i - \tilde{\boldsymbol{y}}_j\|_2$ is always non-negative and therefore $\beta$ also has to be non-negative as it is used to separate links (i.e., $\beta > \|\tilde{\boldsymbol{x}}_i - \tilde{\boldsymbol{y}}_j\|_2$) from non-links (i.e., $\beta < \|\tilde{\boldsymbol{x}}_i - \tilde{\boldsymbol{y}}_j\|_2$). Since both these terms are positive, we can express an exact metric embedding equivalently in squared terms:

$$\text{sign}\left(\beta - \|\tilde{\boldsymbol{x}}_i - \tilde{\boldsymbol{y}}_j\|_2\right) = \text{sign}\left(\beta^2 - \|\tilde{\boldsymbol{x}}_i - \tilde{\boldsymbol{y}}_j\|_2^2\right) = \text{sign}\left(\beta^2 - \left(1 + 1 - 2\tilde{\boldsymbol{x}}_i\tilde{\boldsymbol{y}}_j^\top\right)\right).$$

Setting $\beta = \sqrt{2}$ simplifies to $\text{sign}\left(2\tilde{\boldsymbol{x}}_i\tilde{\boldsymbol{y}}_j^\top\right)$ and since multiplying by 2 does not change the sign, we confirm that the metric embedding can represent the same information as the LPCA-embedding, i.e. $\text{sign}\left(\beta - \|\tilde{\boldsymbol{x}}_i - \tilde{\boldsymbol{y}}_j\|_2\right) = \text{sign}\left(\tilde{\boldsymbol{x}}_i\tilde{\boldsymbol{y}}_j^\top\right).$

As we can write the metric model through an LPCA model augmented with two extra dimensions:

$$\beta^2 - \left( \|\boldsymbol{x}_i\|_2^2 + \|\boldsymbol{y}_j\|_2^2 - 2\boldsymbol{x}_i \boldsymbol{y}_j^\top \right) = \begin{bmatrix} -1 & \|\boldsymbol{y}_j\|_2^2 & \boldsymbol{x}_i \end{bmatrix} \begin{bmatrix} \|\boldsymbol{x}_i\|_2^2 - \beta^2 \\ -1 \\ 2\boldsymbol{y}_j^\top \end{bmatrix},$$

this demonstrates that the LPCA-embedding needs at least two more dimensions to capture the same information as the metric embedding. □

### A.3 Experiments including self-links

The experiments in Chanpuriya et al. (2020) were based on self-loops and for direct comparison we therefore here also include the corresponding experiments in which we set the diagonal of all networks to one (i.e., include self-loops). 10 experiments were run and the results are reported in Table 5. The hyperparameters for the experiment are reported in Table 6.

Table 5: Lowest exact embedding dimensions ($D^*$) found for each dataset across 10 searches along with the mean and standard deviations across the searches. Self-loops are considered. Note that in Chanpuriya et al. (2020) all their graphs were converted to undirected graphs and the rank comparison is therefore not directly comparable for the directed networks. See Figure 1 for graph statistics for each of the graphs.

| Dataset | $D^*$ ($L_2$) | | $D^*$ (LPCA) | | $D^*$ (Chanpuriya et al., 2020) |
|---|---|---|---|---|---|
| Cora | **6** | (6.1 $\sigma$ 0.3162) | 9 | (9 $\sigma$ 0) | 16 |
| Citeseer | **7** | (7 $\sigma$ 0) | 9 | (9.556 $\sigma$ 0.526) | 16 |
| Facebook | **20** | (20 $\sigma$ 0) | 21 | (22.3 $\sigma$ 1.16) | - |
| ca-GrQc | **8** | (8 $\sigma$ 0) | 9 | (10.2 $\sigma$ 0.7888) | 16 |
| Wiki-Vote | **42** | (42.8 $\sigma$ 0.4216) | **42** | (44.1 $\sigma$ 1.449) | 48 |
| p2p-Gnutella04 | **16** | (16 $\sigma$ 0) | 17 | (17.7 $\sigma$ 0.483) | 32 |
| ca-HepPh | **17** | (17.1 $\sigma$ 0.3162) | 19 | (18.7 $\sigma$ 0.483) | 32 |
| Pubmed | **15** | (15 $\sigma$ 0) | 16 | (17.4 $\sigma$ 0.6992) | 48 |

Table 6: Hyperparameters for exact embedding search. 10 experiments were carried out for both the LPCA and the $L_2$ model.

| Dataset | # epochs | Initial learning rate | LR-scheduler patience ($k$) | Search range |
|---|---|---|---|---|
| Cora | 30,000 | 1.0 | 500 | [1, 50] |
| Citeseer | 30,000 | 1.0 | 500 | [1, 50] |
| Facebook | 30,000 | 1.0 | 500 | [1, 50] |
| ca-GrQc | 30,000 | 1.0 | 500 | [1, 50] |
| Wiki-Vote | 30,000 | 1.0 | 500 | [1, 50] |
| p2p-Gnutella04 | 20,000 | 0.1 | 500 | [1, 50] |
| ca-HepPh | 20,000 | 1.0 | 500 | [1, 50] |
| Pubmed | 20,000 | 1.0 | 500 | [1, 50] |

### A.4 Hyperparameters used for results without self-links

The hyperparameters used for the experiments not considering self-links (Figure 2) are presented in Table 7.

### A.5 The advantage of initializing the embedding space by SVD

We used a low-rank SVD of a higher-rank embedding space to initialize the search of the embedding space at the next rank in the Algorithm 1. In Figure 7 we demonstrate the difference between this "hot start" approach and a randomly initialized embedding space i.e. "cold start".

Table 7: Hyperparameters for exact embedding search not considering self-links (see results in Figure 2). 5 experiments were carried out for both the LPCA and the $L_2$ model.

| Dataset | # epochs | Initial learning rate | LR-scheduler patience ($k$) | Search range |
|---|---|---|---|---|
| Cora | $30,000$ | $1.0$ | $200$ | $[1, 64]$ |
| Citeseer | $30,000$ | $1.0$ | $200$ | $[1, 64]$ |
| Facebook | $30,000$ | $1.0$ | $200$ | $[1, 96]$ |
| ca-GrQc | $30,000$ | $1.0$ | $200$ | $[1, 80]$ |
| Wiki-Vote | $30,000$ | $1.0$ | $200$ | $[1, 50]$ |
| p2p-Gnutella04 | $30,000$ | $0.1$ | $200$ | $[1, 80]$ |
| ca-HepPh | $30,000$ | $1.0$ | $200$ | $[1, 80]$ |
| Pubmed | $30,000$ | $1.0$ | $200$ | $[1, 80]$ |

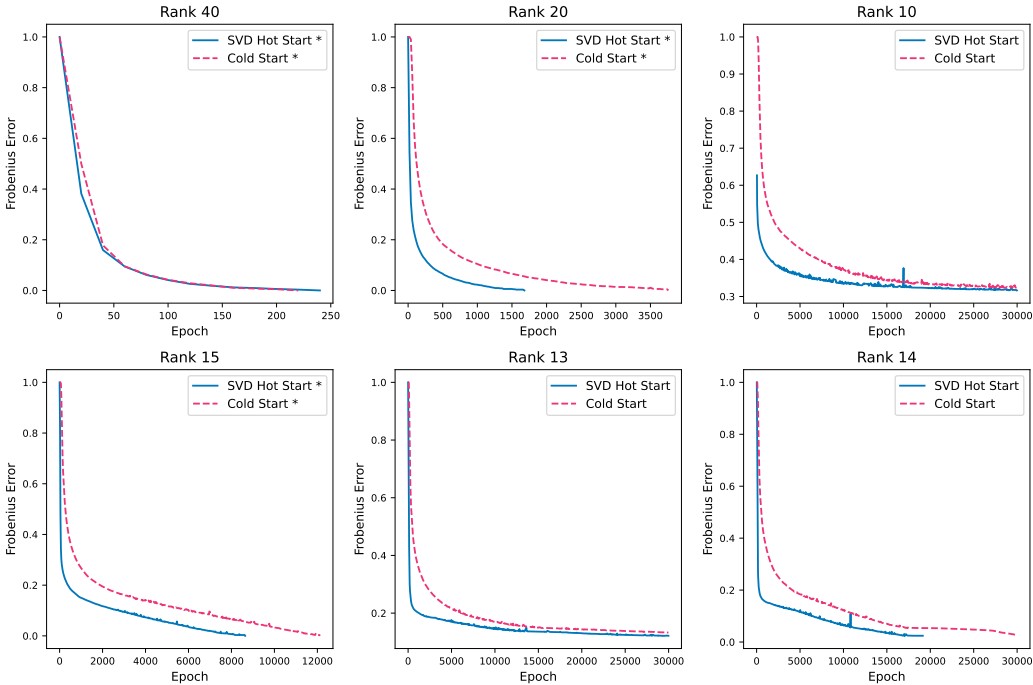

Figure 7: SVD and random initialization (cold start) for some search steps for the dataset PubMed. The search was initialized at rank 80 (randomly initialized) and then low-rank SVD-initialization was used for the proceeding search steps. Additionally, we have initialized training with random initialization for each search step. The legend is marked with $*$ if an exact embedding was achieved.

## A.6 VISUALIZATION OF TRAINING STATISTICS ON SYNTHETIC GRAPHS

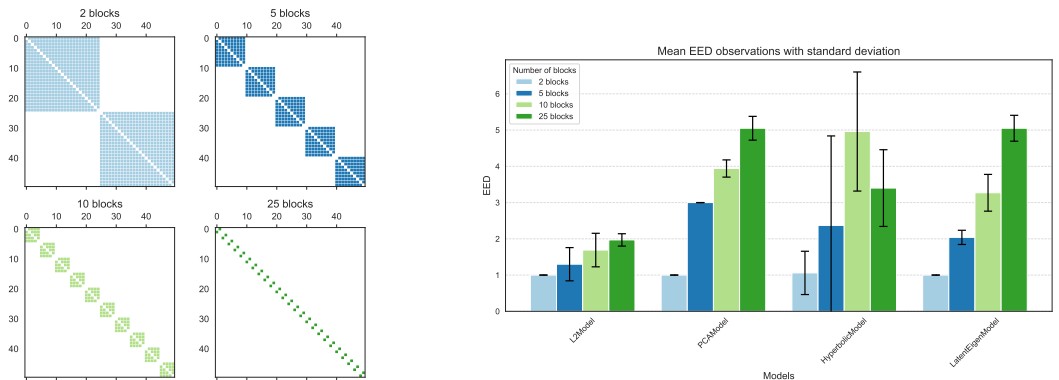

Figure 8: Visualization of the training statistics over 100 test runs on the synthetic graphs seen in the left figure. The bar is the mean exact embedding dimension (EED) and the error bars correspond to the standard deviation of the measurements. All runs are performed using hinge loss.

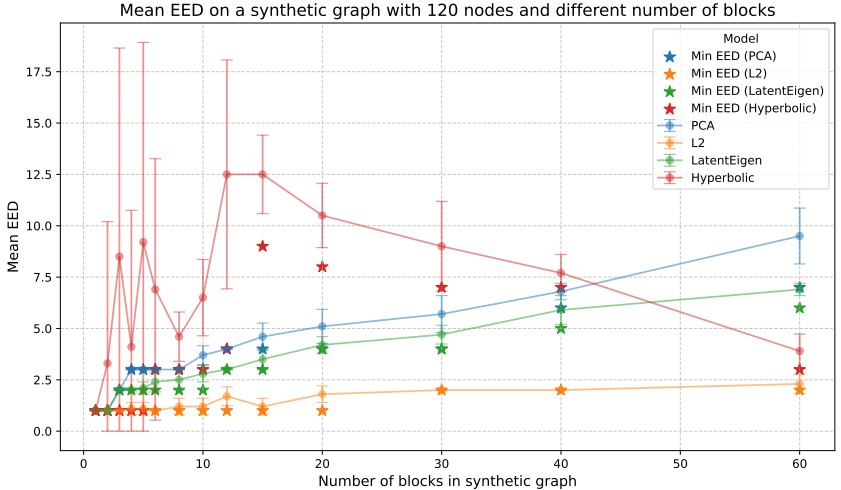

Figure 9: EED for different models and different block sizes in synthetic networks with different amounts of blocks similar to those in Figure 8.

## A.7 THE HIERARCHICAL BLOCK DISTANCE MODEL

Given a Bernoulli log-likelihood over a latent distance model, where $Y_{N \times N} = (y_{i,j}) \in {0, 1}^{N \times N}$ represents the adjacency matrix of the network, where $y_{i,j} = 1$ if the pair $(i, j) \in E$, otherwise it is 0 for all $1 \leq i, j \leq N$. The total network log-likelihood is defined as:

$$\log P(Y|\mathbf{\Lambda}) = \sum_{i \neq j} \left( y_{ij} \eta_{ij} - \log(1 + \exp(\eta_{ij})) \right)$$
$$= \sum_{i \neq j : y_{ij}=1} \eta_{ij} \ - \sum_{i \neq j} \log(1 + \exp(\eta_{ij})), \tag{10}$$

where $\eta_{ij}$ are the log-odds, $\eta_{ij} = \beta - \|\boldsymbol{x}_i - \boldsymbol{y}_j\|_2$. Scaling the optimization of such an expression is not feasible as the second term requires the computation of all node pairs scaling as $\mathcal{O}(N^2)$. In

contrast, the first term scale with the number of edges, i.e. $\mathcal{O}(N \log N)$ Nakis et al. (2022) making its computation scalable for large networks.

To scale the LDM to large-scale networks, the HBDM procedure approximates $\sum_{i \neq j} \log(1 + \exp(\eta_{ij}))$ by enforcing a hierarchical block structure similar to stochastic block models. Specifically, HBDM employs a hierarchical divisive clustering procedure. It is worth noting that the original work by Nakis et al. [2022] focused on a Poisson likelihood and undirected networks. In this study, we extend their approach to a Bernoulli likelihood, adapting it for directed networks.

We define the total representation matrix $\mathbf{Z} = [\mathbf{X}; \mathbf{Y}]$, which combines the source and target node embeddings through concatenation. The matrix $\mathbf{Z}$ is then structured into a hierarchy using a tree-based divisive clustering approach, resulting in a cluster dendrogram.

The tree's root represents a single cluster encompassing the entire set of the concatenated latent variable embeddings, $\mathbf{Z}$. At each level of the tree, the cluster is recursively partitioned until the leaf nodes contain no more than the desired number of nodes, $N_{leaf}$. The number of leaf nodes, $N_{\text{leaf}}$, is chosen based on the HBDMs' linearithmic complexity upper bound and thus set to $N_{\text{leaf}} = \log N$. This results in approximately $K = N/\log(N)$ total clusters. At each tree level, the tree-nodes represent clusters corresponding to that level's height. When partitioning a non-leaf node, the split is performed only on the set of data points assigned to the parent tree-node (cluster). For each tree level, the pairwise distances between data points in different tree-nodes are used to define the distances between the corresponding cluster centroids. These distances are then used to compute the likelihood contribution of the blocks. Binary splits are applied iteratively to the non-leaf tree-nodes, progressing down the tree. When all tree nodes are treated becomes leaves (contain maximum $\log N$ points), the HBDM analytically computes the pairwise distances within each cluster to determine the likelihood contribution of the corresponding analytical blocks. This computation incurs a linearithmic cost of $\mathcal{O}(K N_{\text{leaf}}^2) = \mathcal{O}(N \log N)$. Moreover, this approach preserves the homophily and transitivity properties of the model, as explicitly shown in the paper.

The total HBDM expression for our approach is defined as:

$$\mathcal{L}_{\text{HBDM}}(R) \triangleq \sum_{\substack{i \neq j \\ y_{i,j}=1}} \left( \beta - ||\mathbf{x}_i - \mathbf{y}_j||_2 \right) - \sum_{k_L=1}^{K_L} \left( \sum_{i,j \in C_{k_L}} \log(1 + \exp(\beta - ||\mathbf{x}_i - \mathbf{y}_j||_2)) \right)$$
$$- \sum_{l=1}^{L} \sum_{k=1}^{K_l} \sum_{k' \neq k}^{K_l} \left( \log(1 + \exp(\beta - ||\boldsymbol{\mu}_k^{(l)} - \boldsymbol{\mu}_{k'}^{(l)}||_2)) \right), \tag{11}$$

where $l \in 1, \ldots, L$ denotes the $l$-th level of the dendrogram, $k_l$ is the cluster index at each tree level, and $\boldsymbol{\mu}_k^{(l)}$ represents the corresponding centroid.

### A.7.1 THE EUCLIDEAN CLUSTERING OF HBDM, RESPECTING METRIC PROPERTIES

Finally, in order for the clustering to preserve the metric properties of the Euclidean space, the HBDM defines a Euclidean version of K-means clustering. The divisive clustering procedure relies on the following Euclidean norm objective:

$$J(\mathbf{r}, \boldsymbol{\mu}) = \sum_{i=1}^{N} \sum_{k=1}^{K} r_{ik} |\mathbf{z}_i - \boldsymbol{\mu}_k|_2, \tag{12}$$

where $k$ denotes the cluster index, $\mathbf{z}i$ is the $i$-th data point, $rik$ is the cluster responsibility or assignment, and $\boldsymbol{\mu}_k$ is the cluster centroid. Since the loss function for K-means with the Euclidean norm does not provide closed-form updates, the proposed method introduces the following auxiliary loss function:

Equation 12 as:

$$J^+(\boldsymbol{\phi}, \mathbf{r}, \boldsymbol{\mu}) = \sum_{i=1}^{N} \sum_{k=1}^{K} r_{ik} \left( \frac{||\mathbf{z}_i - \boldsymbol{\mu}_k||_2^2}{2\phi_{ik}} + \frac{1}{2}\phi_{ik} \right), \tag{13}$$

where $\phi$ are the auxiliary variables, while in Nakis et al. (2022) they show how this auxiliary functions accounts for optimizing Equation 12.

## A.8   HBDM SEARCH

The active set used in the hinge loss optimization is reduced significantly by using the solution obtained from the HBDM framework as described in Nakis et al. (2022) and adapted to binary graphs. This reduced active set is then optimized by the hinge loss with a margin of 0. A KDTree is used to query neighbors within the radius of $\beta$ and this is used to update the active set and thus determine which nodes are reconstructed correctly. From Figure 10 we observe empirically that the HBDM has reduced the active set enough to optimize using the hinge loss and we further observe that the next search step hot started with the low-rank SVD of the previous full reconstruction also starts at a sufficiently small active set size.

## A.9   COMPUTE RESOURCES USED

The experiments on the smaller graphs, i.e., all the graphs except com-amazon and roadNetPA can all be run on consumer-grade hardware. More specifically we used a 2023 Macbook Pro with an M3 Pro chip. For the larger datasets we used a high-performance compute cluster equipped with an Nvidia A100 GPU and multiple Intel Xeon Gold CPUs with either 16 or 24 cores.

## A.10   ADDITIONAL STATISTICS OF THE RECONSTRUCTED GRAPH FOR EMBEDDING DIMENSIONS BELOW THE OPTIMAL $D^*$:

We provide pairwise comparisons between the optimal dimension $(D^*)$ and lower dimensions for (i) Degree Distribution, (ii) Clustering Coefficient Distribution, and (iii) Shortest Path Length Distribution. Figure Figure 11 shows results for the Cora dataset while Figure Figure 12 the results for the Facebook dataset, highlighting increasing deviations in distributions as dimensions move further below $(D^*)$.

## A.11   PERFECTLY RECONSTRUCTING RANDOM NETWORKS:

We investigate whether our proposed approach can perfectly reconstruct random networks with a ground-truth latent dimension, for both Euclidean and Hyperbolic geometries. To generate random graphs, we sample $D$-dimensional random vectors uniformly within a ball of radius $R$ (with $R = 1$ for the Poincaré disk model) and assign a scalar bias to ensure realistic network sparsity. Links are then generated via Bernoulli sampling. Our findings indicate that stochastic network generation prevents the recovery of exact embedding dimensions (EED), even in high-dimensional settings. However, by making network generation deterministic — linking nodes when $d_{ij} \leq \beta$ — perfect reconstruction is achieved. We set the number of nodes for all networks as $N = 1000$, and $D = 3, 8$. The results are summarized in Table 8.

Table 8: Comparison between EED found by the hyperbolic model and the Euclidean model for synthetic networks generated according to 3- and 8-dimensional hyperbolic and Euclidean geometric structures, respectively.

| Synthetic Dataset | $D^*$ (Hyperbolic Model) | $D^*$ (Euclidean Model) |
|---|---|---|
| Hyperbolic 3D | 5 | 4 |
| Hyperbolic 8D | 9 | 9 |
| Euclidean   3D | 3 | 3 |
| Euclidean   8D | 8 | 8 |

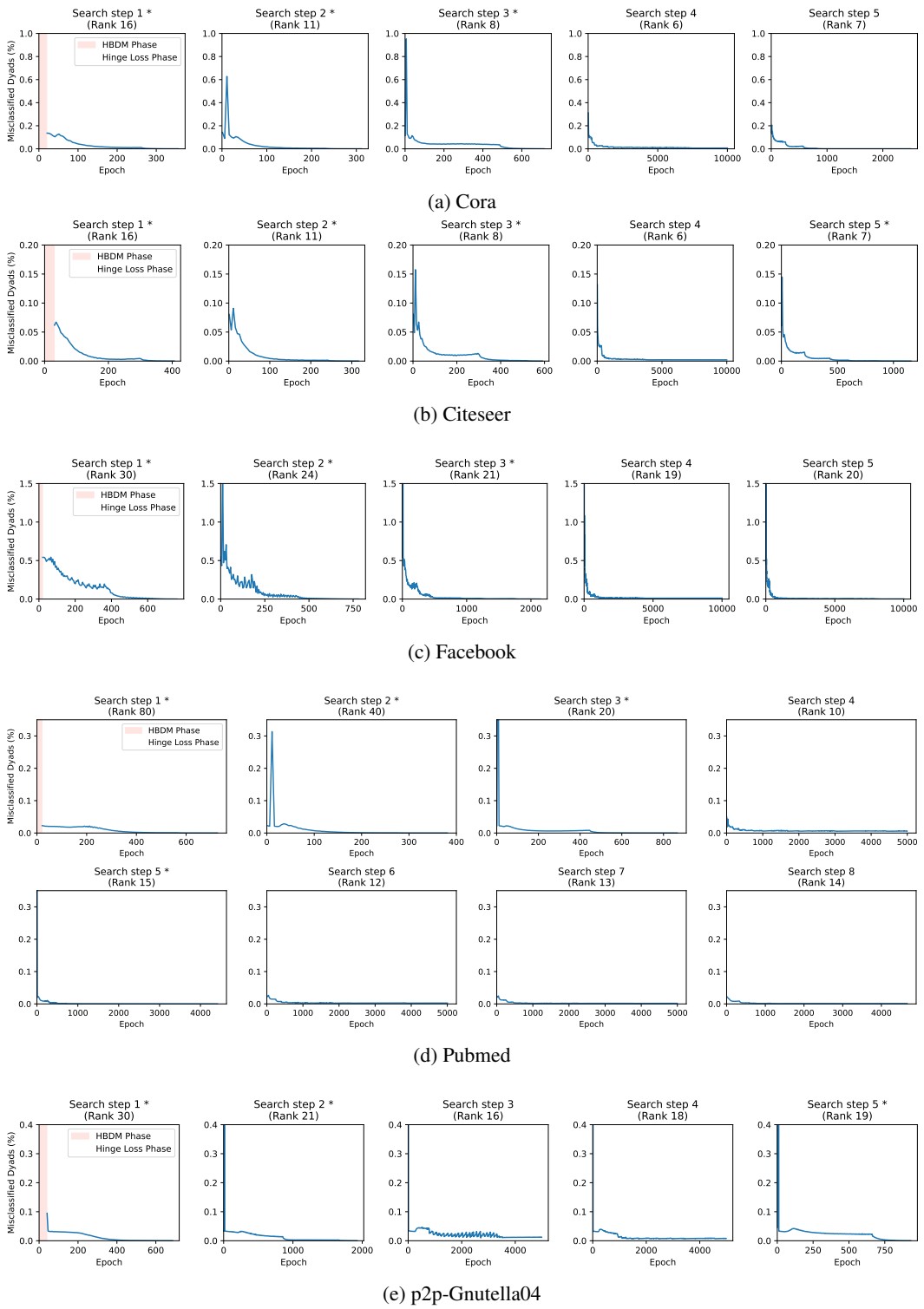

Figure 10: Percentage of misclassified dyads for different steps of the search algorithm using HBDM. The run is marked by a ∗ if an exact embedding was achieved.

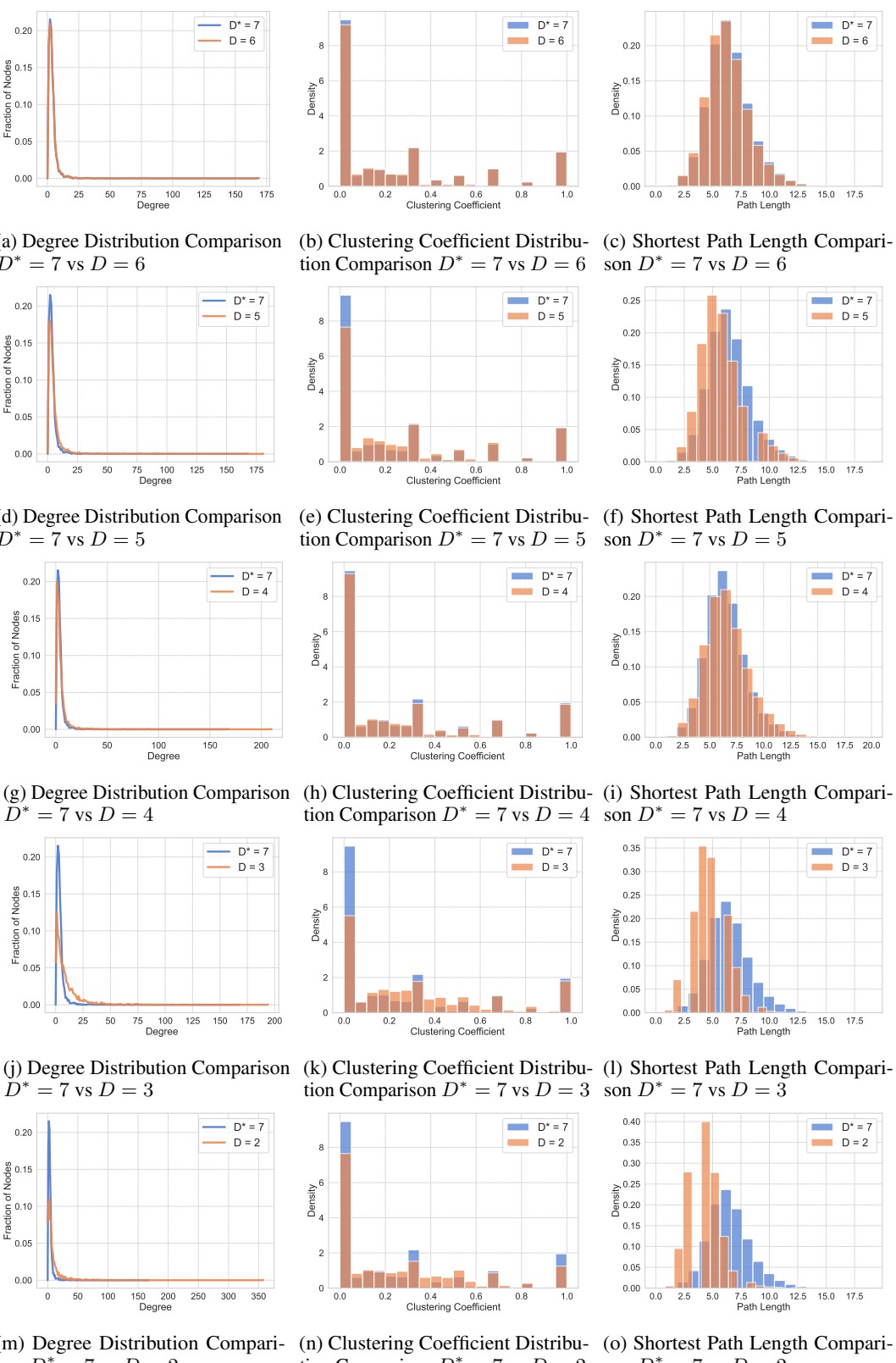

(a) Degree Distribution Comparison $D^* = 7$ vs $D = 6$

(b) Clustering Coefficient Distribution Comparison $D^* = 7$ vs $D = 6$

(c) Shortest Path Length Comparison $D^* = 7$ vs $D = 6$

(d) Degree Distribution Comparison $D^* = 7$ vs $D = 5$

(e) Clustering Coefficient Distribution Comparison $D^* = 7$ vs $D = 5$

(f) Shortest Path Length Comparison $D^* = 7$ vs $D = 5$

(g) Degree Distribution Comparison $D^* = 7$ vs $D = 4$

(h) Clustering Coefficient Distribution Comparison $D^* = 7$ vs $D = 4$

(i) Shortest Path Length Comparison $D^* = 7$ vs $D = 4$

(j) Degree Distribution Comparison $D^* = 7$ vs $D = 3$

(k) Clustering Coefficient Distribution Comparison $D^* = 7$ vs $D = 3$

(l) Shortest Path Length Comparison $D^* = 7$ vs $D = 3$

(m) Degree Distribution Comparison $D^* = 7$ vs $D = 2$

(n) Clustering Coefficient Distribution Comparison $D^* = 7$ vs $D = 2$

(o) Shortest Path Length Comparison $D^* = 7$ vs $D = 2$

Figure 11: Additional pairwise graph statistics comparison for the reconstructed graph as the latent dimension decreases from the exact embedding dimension $(D^*)$ for the Cora network. $(D^*)$ ensures perfect reconstruction.

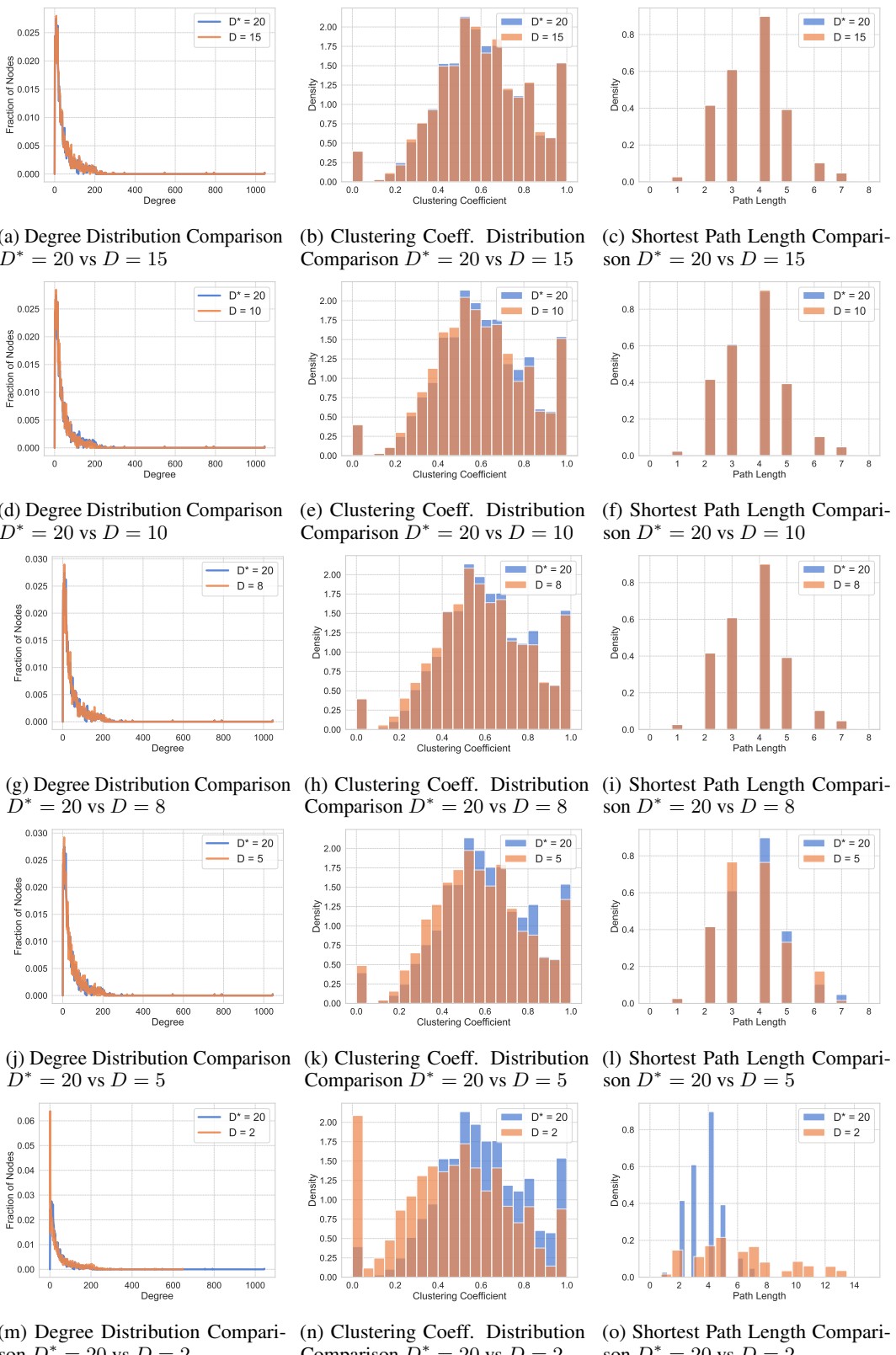

(a) Degree Distribution Comparison $D^* = 20$ vs $D = 15$

(b) Clustering Coeff. Distribution Comparison $D^* = 20$ vs $D = 15$

(c) Shortest Path Length Comparison $D^* = 20$ vs $D = 15$

(d) Degree Distribution Comparison $D^* = 20$ vs $D = 10$

(e) Clustering Coeff. Distribution Comparison $D^* = 20$ vs $D = 10$

(f) Shortest Path Length Comparison $D^* = 20$ vs $D = 10$

(g) Degree Distribution Comparison $D^* = 20$ vs $D = 8$

(h) Clustering Coeff. Distribution Comparison $D^* = 20$ vs $D = 8$

(i) Shortest Path Length Comparison $D^* = 20$ vs $D = 8$

(j) Degree Distribution Comparison $D^* = 20$ vs $D = 5$

(k) Clustering Coeff. Distribution Comparison $D^* = 20$ vs $D = 5$

(l) Shortest Path Length Comparison $D^* = 20$ vs $D = 5$

(m) Degree Distribution Comparison $D^* = 20$ vs $D = 2$

(n) Clustering Coeff. Distribution Comparison $D^* = 20$ vs $D = 2$

(o) Shortest Path Length Comparison $D^* = 20$ vs $D = 2$

Figure 12: Additional pairwise graph statistics comparison for the reconstructed graph as the latent dimension decreases from the exact embedding dimension $(D^*)$ for the Facebook network. $(D^*)$ ensures perfect reconstruction.

