# OpenReview forum: "How Low Can You Go? Searching for the Intrinsic Dimensionality of Complex Networks using Metric Node Embeddings"
_ICLR.cc/2025/Conference — ICLR 2025 Poster_

### Official Review · Reviewer_bMSk · 2024-10-27

**Soundness:** 3
**Presentation:** 2
**Contribution:** 3
**Rating:** 6
**Confidence:** 3

**Summary:**

This paper advances prior work in the field about finding exact representations of networks using low-dimensional vector embeddings in a latent space. It does so by proposing a few algorithms that scale the work to larger networks. First, the paper uses a distance-based model for network representation, which is slightly different from the bilinear model used in other works. The results show that the the former is at least as effective as the latter, and it can also accommodate scalable checking that the representation is exact. Second, the paper finds tighter upper bounds for the dimensionality needed for exact representation than prior work, proposing binary search and warm starts to speed up the procedure. Third, the paper explores using sampling and hierarchical models to facilitate scaling up the search for exact embeddings to large networks, and it claims to find exact representations for a 1M-node network.

**Strengths:**

- The paper advances understanding of the capacity of dense vector representations of a network's nodes. Such representations are ubiquitous across graph machine learning, but this area is still not well understood, so advancements can be significant.
- The discussion of motivation and presentation of results is generally clear.
- The core advancement of using a distance-based model, KD-trees, and sampling and/or hierarchical models to scale up exact embedding search to large graphs seems logical and yields results on significantly larger graphs than prior work.
- Finding tighter upper bounds for exact embedding dimensionality using binary search and warm starts is a smaller but still nice advancement.

**Weaknesses:**

- The adaptation of HBDM is the best instance of this paper's main advancement, which is scaling exact representation search to larger networks. Given this, HBDM is insufficiently described in the main paper. In particular, it is the reviewer's opinion that the discussion of Equation 6 should be significantly expanded with full description of the notation and more intuition about the terms of the loss.
- On the other hand, there is a lot of description of relatively minor differences, e.g., the single dimension of difference between LPCA and the latent eigenmodel, which could be moved to the appendix.
- There could be some more discussion of older but still relevant theory work, such as work on sign rank of a matrix (e.g., Razborov and Sherstov, 2010), since this line of work is essentially about the sign ranks of networks' adjacency matrices.
---
Razborov, Alexander A., and Alexander A. Sherstov. "The sign-rank of AC ^0." SIAM Journal on Computing 39.5 (2010): 1833-1855.

**Questions:**

- Related to the main weakness listed above, how would you intuitively describe Equation 6 and the adaptation of HBDM to exact representation search here?
- Is the KD-tree-based check that the representation is exact used only for the distance-based model (L2), or can is it also applied to LPCA and Eigenmodel? It is not clear to me how it can be applied to the latter two.
- It seems that CC sampling would be overall stronger than RN sampling, but based on Figure 3 it seems to take longer to produce an exact representation. Why is this the case?

### Typos
- Line 136 " yet the same embedding dimension for networks exhibiting heterophily."
- Line 153 "missclassified"
- Line 902 "miss-classify"
- In equation 2, $\mathcal{R}$ is a matrix, but in equation 4, it is a scalar.

---

> ### Author Response · Authors · 2024-11-22
>
> We sincerely thank the reviewer for their constructive feedback and insightful suggestions, which have greatly enhanced our work. Below, we provide detailed responses to each of the points raised.
>
> - The adaptation of HBDM is the best instance of this paper's main advancement, which is scaling exact representation search to larger networks. Given this, HBDM is insufficiently described in the main paper. In particular, it is the reviewer's opinion that the discussion of Equation 6 should be significantly expanded with full description of the notation and more intuition about the terms of the loss.
>
> We appreciate the reviewer’s feedback and have addressed this by incorporating an expanded description of the HBDM procedure in the revised manuscript. Additionally, we have clarified the notation to ensure greater transparency and ease of understanding for readers. This revision aims to provide a more comprehensive and explicit explanation, enabling readers to follow the methodology more effectively and understand its nuances. We hope these enhancements meet the reviewer’s expectations and improve the overall clarity of the manuscript.
>
> - On the other hand, there is a lot of description of relatively minor differences, e.g., the single dimension of difference between LPCA and the latent eigenmodel, which could be moved to the appendix.
>
> Given the one-page extra allowance for the final version of the paper, we plan to retain the description in the manuscript as it is crucial for providing context and clarity to our contributions. Removing or significantly reducing this section would risk oversimplifying important details and potentially hindering the reader's understanding. Instead, we will ensure the additional space is utilized efficiently to further enhance the manuscript, including addressing reviewer feedback and refining our explanations. This approach aligns with our goal of delivering a comprehensive and well-rounded final version.
>
> - There could be some more discussion of older but still relevant theory work, such as work on sign rank of a matrix (e.g., Razborov and Sherstov, 2010), since this line of work is essentially about the sign ranks of networks' adjacency matrices.
>
> We have included a discussion of the sign-rank and its relation to equation 1, as we agree it is a highly relevant point raised by the reviewer. This includes the original lower bound on the sign-rank given by:
>
> Jürgen Forster. A linear lower bound on the unbounded error probabilistic communication complexity. Journal of Computer and System Sciences, 65(4):612–625, 2002.
>
> as well as improved bound in the context of learning theory and communication complexity of:
>
> Alexander A Razborov and Alexander A Sherstov. The sign-rank of ac ˆ0. SIAM Journal on Computing, 39(5):1833–1855, 2010.
>
> and additionally the recent survey given on existing bounds for sign-rank and their limitations:
>
> Hamed Hatami, Pooya Hatami, William Pires, Ran Tao, and Rosie Zhao. Lower bound methods for sign-rank and their limitations. In Approximation, Randomization, and Combinatorial Optimization. Algorithms and Techniques (APPROX/RANDOM 2022). Schloss-Dagstuhl-Leibniz Zentrum für Informatik, 2022
>
> - Is the KD-tree-based check that the representation is exact used only for the distance-based model (L2), or can is it also applied to LPCA and Eigenmodel? It is not clear to me how it can be applied to the latter two.
>
> As the reviewer correctly observed, KD-tree-based check representation is not feasible for the LPCA and Eigenmodel specifications. This limitation highlights a unique strength of leveraging an embedding metric space, which enables efficient and scalable representation methods unavailable in these alternative approaches.
>
> - It seems that CC sampling would be overall stronger than RN sampling, but based on Figure 3 it seems to take longer to produce an exact representation. Why is this the case?
>
> Whereas CC is widely used and in general strong we observe that it exhibits initial fast convergence but then slowly converges in getting the last miss-classified dyads correct. We attribute this to the RN procedure systematically visiting entire blocks of the graphs and thus miss-classified non-link dyads potentially more frequently than when sampling randomly control samples for the non-link dyads.

---

> > ### Comment · Reviewer_bMSk · 2024-11-26
> >
> > Thank you for your response, and for the addition of the related work regarding sign-rank. I have read the added content on HBDM, but still do not believe it defines all the terms and explains the concept completely, so I think the presentation could be improved. I will keep my positive rating.

---

> ### Author Response · Authors · 2024-11-26
> **Answer to Reviewer bMSk 1/2**
>
> We would like to thank the reviewer for their valuable feedback on our updated paper and for providing us with the opportunity to further improve the presentation and clarity of our manuscript. In response, we provide a detailed description of the HBDM approach adopted in our paper. This description has also been included in the supplementary materials (subsection A.7), and we plan to integrate it into the final manuscript as well.
>
>
> ## The Hierarchical Block Distance Model
> Given a Bernoulli log-likelihood over a latent distance model, where $Y_{N \times N} = \left( y_{i,j} \right) \in {0,1}^{N \times N}$ represents the adjacency matrix of the network, where $y_{i,j} = 1$ if the pair $(i,j) \in E$, otherwise it is $0$ for all $1 \leq i , j \leq N$. The total network log-likelihood is defined as:
>
> \begin{align}
>     \log P(Y|\mathbf{H})& =\sum_{i\neq j}\Big(y_{ij}\eta_{ij}-\log(1+\exp{(\eta_{ij})})\Big)\nonumber
> \\
> & =\sum_{i\neq j:y_{ij}=1}\eta_{ij} - \sum_{i\neq j}\log(1+\exp{(\eta_{ij})}),
> \end{align}
>
> where $\eta_{ij}$ are the log-odds, $\eta_{ij}=\beta -|\vec{x}_i - \vec{y}_j|_2$. Scaling the optimization of such an expression is not feasible as the second term requires the computation of all node pairs scaling as $\mathcal{O}(N^2)$. In contrast, the first term scale with the number of edges, i.e. $\mathcal{O}(N\log N)$ [Nakis et al. 2022] making its computation scalable for large networks.
>
> To scale the LDM to large-scale networks, the HBDM procedure approximates $\sum_{i \neq j} \log(1 + \exp(\eta_{ij}))$ by enforcing a hierarchical block structure similar to stochastic block models. Specifically, HBDM employs a hierarchical divisive clustering procedure. It is worth noting that the original work by Nakis et al. [2022] focused on a Poisson likelihood and undirected networks. In this study, we extend their approach to a Bernoulli likelihood, adapting it for directed networks.
>
> We define the total representation matrix $\mathbf{Z} = [\mathbf{X}; \mathbf{Y}]$, which combines the source and target node embeddings through concatenation. The matrix $\mathbf{Z}$ is then structured into a hierarchy using a tree-based divisive clustering approach, resulting in a cluster dendrogram.
>
> The tree's root represents a single cluster encompassing the entire set of the concatenated latent variable embeddings, $\mathbf{Z}$. At each level of the tree, the cluster is recursively partitioned until the leaf nodes contain no more than the desired number of nodes, $N_{leaf}$. The number of leaf nodes, $N_{\text{leaf}}$, is chosen based on the HBDMs' linearithmic complexity upper bound and thus set to $N_{\text{leaf}} = \log N$. This results in approximately $K = N / \log(N)$ total clusters. At each tree level, the tree-nodes represent clusters corresponding to that level’s height. When partitioning a non-leaf node, the split is performed only on the set of data points assigned to the parent tree-node (cluster). For each tree level, the pairwise distances between data points in different tree-nodes are used to define the distances between the corresponding cluster centroids. These distances are then used to compute the likelihood contribution of the blocks. Binary splits are applied iteratively to the non-leaf tree-nodes, progressing down the tree. When all tree nodes become leaves (containing maximum $\log N$ points), the HBDM analytically computes the pairwise distances within each cluster to determine the likelihood contribution of the corresponding analytical blocks. This computation incurs a linearithmic cost of $\mathcal{O}(K N_{\text{leaf}}^2) = \mathcal{O}(N \log N)$. Moreover, this approach preserves the homophily and transitivity properties of the model, as explicitly  shown in the paper.
>
> The total HBDM expression for our approach is defined in Equation 6 of the main paper, where $l \in {1, \dots, L}$ denotes the $l$-th level of the dendrogram, $k_l$ is the cluster index at each tree level, and $\mathbf{\mu}_k^{(l)}$ represents the corresponding centroid.

---

> > ### Author Response · Authors · 2024-11-26
> > **Answer to Reviewer bMSk 2/2**
> >
> > ## The Euclidean Clustering in HBDM, respects the metric properties our model.
> >
> >
> > In order for the clustering to preserve the metric properties of the Euclidean space, the HBDM defines a Euclidean version of K-means clustering. The divisive clustering procedure relies on the following Euclidean norm objective:
> >
> > \begin{equation}
> > J(\mathbf{r}, \mathbf{\mu}) =
> > \sum_{i=1}^N \sum_{k=1}^K r_{ik} \|\mathbf{z}_i - \mathbf{\mu}_k\|_2
> > \end{equation}
> >
> > where $k$ denotes the cluster index, $\mathbf{z}i$ is the $i$-th data point, $r{ik}$ is the cluster responsibility or assignment, and $\mathbf{\mu}_k$ is the cluster centroid. Since the loss function for K-means with the Euclidean norm does not provide closed-form updates, the proposed method introduces an auxiliary loss function  [ Nakis et al. 2022] accounting for optimizing $J(\mathbf{r}, \mathbf{\mu})$.

---

### Official Review · Reviewer_N48b · 2024-10-31

**Soundness:** 3
**Presentation:** 2
**Contribution:** 3
**Rating:** 6
**Confidence:** 4

**Summary:**

The authors investigate how many embedding dimensions are required to losslessly embed complex networks in metric spaces. As a motivation for their work, they mention the importance of low-dimensional embeddings for tasks "such as node classification, link prediction, community detection, network visualization, and network compression". Specifically, they ask whether lower-dimensional embeddings than previously achieved based on the so-called Logistic PCA model are feasible. They develop an efficient algorithm to compute the exact embedding dimension of large complex networks which is based on the hierarchical block distance model. Using kd-trees to efficiently search for nodes' neighbours during optimisation, they avoid $O(n^2)$ complexity, enabling computing lossless embeddings in time $O(n \log n)$. Finally, they show empirically that their approach requires fewer dimensions to embed networks than previous approaches.

**Strengths:**

1. The authors consider a relevant research question and improve upon the state-of-the-art, achieving network embeddings in lower-dimensional metric spaces than previous works.
2. Their proposed $O(n \log n)$ time algorithm for computing exact embeddings in is well-described and easy enough to follow.
3. The empirical evaluation supports their theoretical claims.

**Weaknesses:**

1. The authors highlight their work's relevance by referring to several downstream tasks which are expected to benefit from lower-dimensional embeddings. However, they merely show that they can achieve lower-dimensional embeddings than previous work, but do not show how or whether the performance on downstream tasks is affected.
2. The last paragraph of the abstract mentions that their work can be used to "guide existing network analysis tasks", however, they do not pick up this point later in the paper to explain _how_ this can be done.
3. Some claims appear to be left somewhat vague (see my questions below).

**Questions:**

There are a few things that remained unclear to me, specifically
1. The present analysis is limited to unweighted networks and only considers the presence or absence of links. Would it be possible to extend the proposed approach to weighted networks or is there a fundamental limitation that prevents this?
2. At the points when Fig. 1 is mentioned in the text (l.237), it was not clear to me how to interpret what is shown in the figure. I assume that the key is what is mentioned in l.285-286, is that right? I would recommend providing an explanation of how to read the figure earlier on.
3. I am also wondering whether you have explored how your method behaves on "difficult" networks with less clear structure than those shown in Fig. 1? I am not sure what exactly "difficult structure" is, but would assume that, for example, Erdős–Rényi random graphs may be difficult to embed since they do not exhibit significant structure.
4. Does the proposed Theorem 2.1 always hold in practice? Are there some assumptions that are made?
5. Why is Algorithm 1 efficient? This is not really clear to me. And what does "efficient" mean here (i.e., what is the algorithm's time complexity)?
6. L. 357 states that a "relatively small number of dyads will remain misclassified". Can you say more precisely what "relatively small" means?
7. L. 368-269 states that "The active set $\mathcal{S}$ is then updated [...] until perfect reconstruction is achieved". I assume here that Eqs. (6) and (7) are optimised in turn, is that correct? And if so, how do we know that the procedure will terminate?
8. I was a bit surprised to see that the Roadnet-PA network requires 16 dimensions for lossless embedding. Given that road networks are essentially embedded in 2D in the real world, should it not be possible to embed it in fewer than 16 dimensions?
9. The network shown in Fig. 2 seems like it should be embeddable in 1D, just as the network shown in Fig. 1a. However, it seems somewhat difficult for the optimisation to find this solution, given that the mean EED is higher than 1. What happens as one increases the number of blocks in the networks? Does the mean EED grow?

Minor points
- I believe there should be a comma instead of a period in l. 68 (20.000 nodes --> 20,000 nodes)
- I believe it should be "coarse" instead of "course" in l. 142/143
- I think it should be "obtainable" instead of "obtain able" in Theorem 2.1.

---

> ### Author Response · Authors · 2024-11-22
> **Rebuttal by Authors 1/2**
>
> We thank the reviewer for their detailed and constructive feedback. Below, we address each point raised.
>
> - The authors highlight their work's relevance by referring to several downstream tasks which are expected to benefit from lower-dimensional ... but do not show how or whether the performance on downstream tasks is affected.
>
> We intentionally excluded downstream tasks from the scope of this paper, as the focus is on theoretical contributions. However, we have expanded Section 4 and included analyses (Figure 4 and Supplementary Figures 10 and 11) demonstrating how reducing embedding dimensions below the exact embedding dimension (EED) affects key network statistics. Additionally, we note that prior work (HBDM) established strong link prediction and node classification performance using the LDM framework, which we have clarified in the revised manuscript.
>
> - The last paragraph of the abstract mentions that their work can be used to "guide existing network analysis tasks", however, they do not pick up this point later in the paper to explain how this can be done.
>
> In this paper, we focus on graph analysis tasks, particularly the characterization of latent structures within networks. Figure 5 illustrates this by providing visualizations of the re-ordered adjacency matrix generated through the HBDM procedure. Additionally, latent distance models are known to perform well in downstream tasks, especially in low-dimensional settings, as demonstrated in LDM works such as the HBDM paper.
>
>
>
> - The present analysis is limited to unweighted networks and only considers the presence or absence of links. Would it be possible to extend the proposed approach to weighted networks or is there a fundamental limitation that prevents this?
>
> Extending the approach to weighted networks introduces challenges due to the requirement for exact reconstruction based on a single threshold, as weights increase the complexity of maintaining reconstruction fidelity. While this is feasible, we highlight the additional computational trade-offs and note that it remains an area for future exploration.
>
>
> - At the points when Fig. 1 is mentioned in the text (l.237), it was not clear to me how to interpret what is shown in the figure. I assume that the key is what is mentioned in l.285-286, is that right? I would recommend providing an explanation of how to read the figure earlier on.
>
> We agree and have added an earlier explanation to help readers interpret Figure 1 more effectively.
>
> - I am also wondering whether you have explored how your method behaves on "difficult" networks with less clear structure than those shown in Fig. 1?
>
> We demonstrate in the supplementary material that deterministic reconstruction of random graphs is possible under binary link probabilities. However, embedding noisy graphs such as Erdős–Rényi graphs remains challenging, as their lack of structure increases reconstruction difficulty.
>
>
> - Does the proposed Theorem 2.1 always hold in practice? Are there some assumptions that are made?
>
> The theorem holds in practice as we also empirically verify in synthetic data. However, a limitation of the Theorem is that it relies on the Euclidean norm and thus may not generalize to other metric embeddings such as hyperbolic embeddings considered in the supplementary material of the  revised manuscript.
>
>
> - Why is Algorithm 1 efficient? This is not really clear to me. And what does "efficient" mean here (i.e., what is the algorithm's time complexity)?
>
> The binary search strategy also called logarithmic search exhibits logarithmic scaling and is therefore efficient in that it reduces a search space of K integer values into a search scaling as log(K) while ensuring the results are equivalent to the exhaustive evaluation of all K integer values.We have clarified this in the paper. Furthermore, we explore a warm start using the truncated SVD and demonstrate its efficiency in Figure 6 of the supplementary material.
>
> - L. 357 states that a "relatively small number of dyads will remain misclassified". Can you say more precisely what "relatively small" means?
>
> The number of misclassified dyads is proportional to cNlog⁡N, where c≪N. This ensures efficient optimization, as detailed in the revised manuscript.
>
> - L. 368-269 states that "The active set is then updated [...] until perfect reconstruction is achieved". I assume here that Eqs. (6) and (7) are optimised in turn, is that correct? And if so, how do we know that the procedure will terminate?
>
> Yes, Eqs. (6) and (7) are optimized sequentially. Termination is guaranteed when no misclassified dyads remain, as confirmed through the hinge-loss optimization process.

---

> > ### Author Response · Authors · 2024-11-22
> > **Rebuttal by Authors 2/2**
> >
> > - I was a bit surprised to see that the Roadnet-PA network requires 16 dimensions for lossless embedding. Given that road networks are essentially embedded in 2D in the real world, should it not be possible to embed it in fewer than 16 dimensions?
> >
> > While road networks are inherently 2D, higher-dimensional embeddings capture latent structural properties, such as connectivity patterns or hierarchies, that go beyond pure spatial representation. These properties likely contribute to the observed dimensionality requirements.
> >
> > - The network shown in Fig. 2 seems like ... in the networks? Does the mean EED grow?
> >
> > We have added experiments in the supplementary material where we vary the block size in networks similar to Fig. 2. These experiments demonstrate how the mean EED evolves as the block size increases, providing insights into the challenges of optimization in such cases. The results and analysis are included in Supplementary Figures 7 and 8.

---

> > > ### Comment · Reviewer_N48b · 2024-11-27
> > >
> > > I would like to thank the authors for providing clear answers to my questions and raise my score accordingly.

---

> > > > ### Author Response · Authors · 2024-11-27
> > > > **Thank you!**
> > > >
> > > > Thank you very much for carefully reading the paper, providing constructive and extensive feedback, and, of course, for raising the score!

---

### Official Review · Reviewer_qGwB · 2024-11-01

**Soundness:** 3
**Presentation:** 3
**Contribution:** 2
**Rating:** 5
**Confidence:** 4

**Summary:**

Built on a series of works by Chanpuriya et al. (see [1] and [2]]), this paper uses metric embedding to study the low intrinsic dimension of network embedding. First of all, the authors provide an improved upper bound of existing intrinsic dimensions and an efficient searching algorithm for finding such an upper bound. Experiments were conducted on real-world graphs to demonstrate by considering metric embeddings using the latent distance model. These findings could help to have a better understanding of network embeddings and potentially can be applicable to large-scale graphs.

[1] Sudhanshu Chanpuriya, Cameron Musco, Konstantinos Sotiropoulos, and Charalampos E. Tsourakakis. Node Embeddings and Exact Low-Rank Representations of Complex Networks. Neural Information Processing Systems, 2020.

[2] Sudhanshu Chanpuriya, Ryan Rossi, Anup B. Rao, Tung Mai, Nedim Lipka, Zhao Song, and Cameron Musco. Exact representation of sparse networks with symmetric nonnegative embeddings. In A. Oh, T. Naumann, A. Globerson, K. Saenko, M. Hardt, and S. Levine (eds.),
Advances in Neural Information Processing Systems, volume 36, pp. 21023–21038. Curran Associates, Inc., 2023

**Strengths:**

1. This paper studies interesting theoretical problems concerning the intrinsic dimension of network embeddings. The authors proposes to use a new loss function to quantify the upper bound of intrinis dimension.

2. An efficient searching algorithm is proposed to quantify the provided upper bound. Experimental results also show the effectiveness of the proposed method.

**Weaknesses:**

1. I am not quite sure how significant such a new upper bound is compared with the existing work presented in Theorem 2.1. It seems that the authors use a new objective defined in Equ. (4) to obtain the upper bound $D_{L2}$ which could be $D_{LPCA}$, $D_{LPCA} -1 $, or $D_{LPCA}-2$. How significant is this result? Please make a short discussion on how significant difference between current work and [1,2].

2. It is a concern that the efficient logarithmic search method presented is not effectively justified.  The authors claimed that they derived an efficient logarithmic search strategy that reliably identified an upper bound on the intrinsic dimensionality of exact network embeddings. I did not see the runtime complexity analysis and justification of correctness. How can we make sure the returned embeddings are exactly correct? How can the lower bound and upper bound parameters lb, ub be set?

3. How practically useful of these quantified embeddings compared with DeepWalk and node2vec? Although this is not a very serious concern as the paper mainly focuses on the theoretical perspective of network embedding. How could usefulness be compared with popular node embedding algorithms such as DeepWalk, node2vec, and LINE etc. From my understanding, these real-world embeddings perform better in terms of node classification when the dimensionality increases. It would be helpful if authors can elaborate on this.

**Questions:**

See the weakness above.

---

> ### Author Response · Authors · 2024-11-22
> **Rebuttal by Authors**
>
> We sincerely thank the reviewer for their detailed and constructive feedback. Below, we address each point raised in detail.
>
> - I am not quite sure how significant such a new upper bound is compared with the existing work presented in Theorem 2.1. It seems that the authors use a new objective defined in Equ. (4) to obtain the upper bound.
>
> Our approach provides a substantial improvement in obtaining an upper bound for the dimensionality required for exact network reconstruction. Table 2 demonstrates that our method identifies significantly lower exact embeddings compared to prior work [1] and [2]. Unlike these works, our use of metric embedding spaces allows for efficient scaling to large networks, leveraging Euclidean distance to naturally capture transitivity properties through the triangle inequality. To the best of our knowledge, our method is the only one capable of finding an upper bound on the exact embedding dimension (EED) for large-scale networks.
>
> - It is a concern that the efficient logarithmic search method presented is not effectively justified. The authors claimed that they derived an efficient logarithmic search strategy that reliably identified an upper bound on the intrinsic dimensionality of exact network embeddings. I did not see the runtime complexity analysis and justification of correctness. How can we make sure the returned embeddings are exactly correct? How can the lower bound and upper bound parameters lb, ub be set?
>
>
> The runtime complexity of our approach follows a logarithmic binary search, which is a well-established method for navigating ordered ranks. We provide justification for the correctness of the embeddings through empirical validation and error bars, ensuring robustness in our results. The parameters lb and ub are initialized based on prior dimensionality estimates (e.g., bounds from maximum degree or arboricity), which are discussed in Section 2.2 of the revised paper. We have expanded the discussion in the paper to clarify these points.
>
>
> - How practically useful are these quantified embeddings compared with DeepWalk and node2vec? How could usefulness be compared with popular node embedding algorithms such as DeepWalk, node2vec, and LINE, etc.?
>
> We have excluded direct comparisons to DeepWalk, node2vec, and LINE from the current paper’s scope, as it focuses on the theoretical aspects of exact network embeddings. However, prior work using the LDM model, such as HBDM, demonstrates superior link prediction and node classification performance in the low-dimensional regime, outperforming DeepWalk, node2vec, and LINE. We highlight this context in Section 4.1 of the revised paper.

---

### Official Review · Reviewer_GFnZ · 2024-11-03

**Soundness:** 2
**Presentation:** 1
**Contribution:** 3
**Rating:** 8
**Confidence:** 3

**Summary:**

This paper explores the intrinsic dimensionality of complex networks using metric node embeddings and argues that metric embeddings are superior to vector-based inner product embeddings in this domain. The premise of this paper is dimension reduction, but it is not mentioned at what cost it is acheived. This paper has theoretical, methodological, and empirical contributions as the following.

a) Theoretical Contribution:
- They proved that metric embeddings can provide lower dimensional representations compared to Logistic PCA.
- They showed that metric embeddings can achieve dimensions up to two dimensions less than LPCA while maintaining perfect reconstruction.

b) Methodological Contributions:
- They developed an efficient search algorithm to find the minimal embedding dimension with O(N log N) complexity.
- For large networks, they Proposed a scalable O(N log N) approach using KD-trees.
- They Introduced a two-phase optimization approach combining the Hierarchical Block Distance Model (HBDM) with hinge-loss optimization

c) Empirical Contributions:
- Using their method, they found lower embedding dimensions than previously reported.
- They showed successful embedding of large-scale networks (up to 1 million nodes).
- They provided extensive experiments on various real-world networks

Overall, this has strong theoretical and practical contributions to network embedding research. The experiments are generally sufficient, although it lacks some runtime reports. The paper's main strengths lie in its theoretical foundations and scalable approach for large networks, while its limitations are mainly related to optimization challenges and the scope of experimental validation.

**Strengths:**

Strengths:
- Strong theoretical foundation with proven bounds
- Novel scalable approach for large networks
- Comprehensive empirical validation
- Clear practical implications for network analysis
- Reproducible research with code provided
- Well-written and structured presentation

Strengths with experimentations:
- Diverse dataset selection (10 different networks of varying sizes and types)
- Multiple experimental validations (5 searches per dataset)
- Comprehensive comparisons with baseline methods
- Clear ablation studies showing the benefits of different components
- Scalability demonstrated on large networks
- Reproducibility ensured with detailed hyperparameter settings

**Weaknesses:**

Weaknesses:
- The approach may not find the absolute lowest possible embedding dimension
- Optimization can be sensitive to initialization and hyperparameters
- Limited exploration of alternative metric spaces (e.g., hyperbolic geometry)
- The method might not be directly applicable for link prediction tasks
- Some privacy concerns regarding potential surveillance applications
- While suggested method has low computational complexity, the proposed implementation for preserving the A matrix in the memory is not efficient. In this work they resort to sampling methods for maintaining large A matrices in the memory.

Some additional experiments could strengthen the paper:
- Comparison with other metric embedding approaches (e.g., hyperbolic embeddings)
- More extensive analysis of the impact of network properties on embedding dimension. Features such as connectivity, number of connected components, etc.
- More detailed runtime analysis for different network sizes

**Questions:**

The optimization is sensitive to initialization and hyperparameters. For a given set of hyperparameters, what is the probability of converging to (1) the global minimum answer, (2) a minimum answer, and (3) not converging? Using a Monte Carlo scheme, what would be average value for D for a given set of hyperparams?

Minor comments and suggestions.
- line 045: "...preserving structure from* the discrete graph..." -> "preserving structure *of the discrete graph"
- line 068: 20.000 nodes -> 20,000 nodes. Use a comma instead of period
- line 076: "...we exploit* that the metric properties...". Use of a better verb is recommended.
- line 078: "...exploit* how hierarchical...". Use of a better verb is recommended.
- line 140: "wrt." -> "w.r.t." Use periods.
- line 263: $\mathbb{R}^{2N×D_0}$ -> $\mathbb{R}^{N×2D_0}$. Since you are defining $\mathbf{Z}=[\mathbf{X} \mathbf{Y}]^T$, it seems the dimensions should be $N×2D_0$  rather than $2N×D_0$.
- line 300: "Everything in the graph is connected"? Do you meant a "fully connected graph"?
- line 303: "Scalable inference:" Are you missing a colon?
- line 446: In Figure 3, not all Y-axis ranges are equal. Is there a reason? The Y-axis range for "Rank 16" is [0, 2.0], while for the others is [0, 1.0].
- line 865: In Figure 5, not all Y-axis ranges are equal. In "Rank 10" the range is [0.3, 1.0], in "Rank 13" the range is [0.2, 1.0], while in the rest it is [0.0, 1.0].

---

> ### Author Response · Authors · 2024-11-22
> **Rebuttal by Authors**
>
> We sincerely thank the reviewer for their constructive feedback and suggestions, which have significantly helped us improve our paper. Below, we address each of the points raised in detail.
>
>
> - The approach may not find the absolute lowest possible embedding dimension.
>
> We agree with this limitation, as also discussed in the "Limitations" section of the paper. To address this, we include error bars in all results to account for the uncertainty in determining the lowest embedding dimension. Additionally, we provide new synthetic results in the supplementary material (Figures 7 and 8).
>
> - Optimization can be sensitive to initialization and hyperparameters.
>
> We agree and have explicitly acknowledged this in the "Limitations" section. To address this, we include error bars reflecting the variability of the results. Furthermore, our supplementary material now contains an analysis of initialization sensitivity using synthetic networks with known dimensionality (Figures 7 and 8), which helps highlight this challenge.
>
> - Limited exploration of alternative metric spaces (e.g., hyperbolic geometry).
>
> We have now included additional results in the supplementary material using hyperbolic geometry, specifically employing the Poincaré distance. These results are discussed in Section 4.1 of the revised paper and presented in supplementary Table 7, along with Figures 7 and 8.
>
>
> - The method might not be directly applicable for link prediction tasks.
>
> We chose to exclude link prediction tasks from the scope of the current paper. However, we note that the HBDM framework demonstrates strong performance for link prediction and node classification tasks in prior work, leveraging low-dimensional embeddings based on the LDM model specification.
>
> - Some privacy concerns regarding potential surveillance applications.
>
> We appreciate this comment and fully agree with the concern. We have expanded the discussion on privacy concerns in the "Broader Impact" section of the revised paper to provide a more detailed analysis of potential surveillance implications.
>
>
> - While the suggested method has low computational complexity, the proposed implementation for preserving the A matrix in the memory is not efficient. In this work, they resort to sampling methods for maintaining large A matrices in the memory.
>
> We clarify that our method only stores edges, which typically scale sublinearly. Starting from the linearithmic initialization based on HBDM, we then focus on a small active set of misclassified dyads for hinge-loss optimization. By leveraging KD-tree searches, this optimization scales with the number of links in the network. These clarifications and additional details are now included in the revised manuscript.
>
>
> - Comparison with other metric embedding approaches (e.g., hyperbolic embeddings).
>
> Following the reviewer’s suggestion, we have included additional results comparing hyperbolic embeddings using the Poincaré distance. These results are presented in supplementary Table 7 and Figures 7 and 8, along with added discussion in Section 4.1.
>
>
>
> - More extensive analysis of the impact of network properties on embedding dimension, such as connectivity, number of connected components, etc.
>
> We have included a detailed analysis of the impact of network properties on embedding dimension in the revised paper. Using the Cora network as an example, we examine how statistics such as degree distribution, clustering coefficient, and giant component size deteriorate as the embedding dimension decreases below the exact embedding dimension. These results are presented in Figure 4 (main paper) and Figures 10 and 11 (supplementary material).
>
>
> Regarding the minor corrections, we thank the reviewer for these helpful corrections, which have been addressed in the revised manuscript.

---

### Official Review · Reviewer_odRC · 2024-11-10

**Soundness:** 2
**Presentation:** 2
**Contribution:** 2
**Rating:** 6
**Confidence:** 5

**Summary:**

The paper is discussing a relevant problem in network geometry, which is about searching for the intrinsic dimensionality of complex networks.
The Authors provide an efficient logarithmic search procedure for identifying the exact embedding dimension. Specifically, they propose an algorithm to efficiently search for an upper bound for the exact embedding dimension (D∗) of graphs. The Authors claim to demonstrate how metric embeddings enable inference of the exact embedding dimensions of large-scale networks by exploiting that the metric properties can
be used to provide linearithmic scaling.

**Strengths:**

The idea proposed is original and the question of the study is of significance.

**Weaknesses:**

The study does not refer to certain relevant literature in complex networks.
The study does not use as benchmark any complex network model that generates networks in a geometric space of a given dimensionality.
The results are of difficult to interpret is the study does not provide tests on artificial network models generated in a geometric space with a certain dimensionality.

**Questions:**

Can the authors offer results and tests using for benchmarks artificial networks generated in a geometric space of a given dimension?
For instance, see this article in case of hyperbolic geometry:
Generalised popularity-similarity optimisation model for growing hyperbolic networks beyond two dimensions
B Kovács, SG Balogh, G Palla
Scientific Reports 12 (1), 968

Can the author expand their method also to network with hyperbolic geometry?
Can the authors comments why and how their method applies to many real networks that have intrinsic hyperbolic geometry considering that their method seems based on an Euclidean embedding?

How this study related to this other study that is missing in their related works part:
Detecting the ultra low dimensionality of real networks
P Almagro, M Boguñá, MÁ Serrano
Nature communications 13 (1), 6096

When the Authors mention in the limitation section the to embedding using hyperbolic geometry, they refer to (Nickel & Kiela, 2017; 2018).

It is not clear to me why they do not refer to:
+ the first model-based algorithms proposed in:
1. Curvature and temperature of complex networks
D Krioukov, F Papadopoulos, A Vahdat, M Boguná
Physical Review E, 2009
2. Sustaining the internet with hyperbolic mapping
M Boguná, F Papadopoulos, D Krioukov
Nature communications 1 (1), 62, 2010
3. Network mapping by replaying hyperbolic growth
F Papadopoulos, C Psomas, D Krioukov
IEEE/ACM Transactions on Networking 23 (1), 198-211, 2015

+ the first model-free and topological machine learning algorithm proposed in:
1. Machine learning meets network science: dimensionality reduction for fast and efficient embedding of networks in the hyperbolic space
Josephine Maria Thomas, Alessandro Muscoloni, Sara Ciucci, Ginestra Bianconi, Carlo Vittorio Cannistraci
https://arxiv.org/abs/1602.06522, 2016
2. Machine learning meets complex networks via coalescent embedding in the hyperbolic space
A Muscoloni, JM Thomas, S Ciucci, G Bianconi, CV Cannistraci
Nature Communications 8 (1), 1, 2017

---

> ### Author Response · Authors · 2024-11-22
> **Rebuttal by Authors**
>
> We sincerely thank the reviewer for their constructive feedback and for bringing to our attention relevant literature we had previously overlooked. Below, we address each point in detail:
>
> - Can the authors offer results and tests using for benchmarks artificial networks generated in a geometric space of a given dimension?
>
> We appreciate the suggestion and have included benchmarks using artificial networks generated in both Euclidean and hyperbolic spaces. Results for these benchmarks are provided in the supplementary material (Subsection A.9 and Table 8), with an emphasis on the Poincaré disk model for hyperbolic geometry.
>
> - Can the author expand their method also to network with hyperbolic geometry?
>
> We have extended our method to networks with hyperbolic geometry and included results using the Poincaré distance for embedding in the supplementary material. This demonstrates the feasibility of applying our approach to hyperbolic settings.
>
>
> - Can the authors comment why and how their method applies to many real networks that have intrinsic hyperbolic geometry considering that their method seems based on an Euclidean embedding?
>
> While our method primarily focuses on Euclidean embeddings, we now include a discussion in the limitations section on the relationship between our approach and networks with intrinsic hyperbolic geometry. Additionally, we provide examples demonstrating that metric-based embeddings in Euclidean spaces can approximate certain hyperbolic network properties under specific conditions.
>
> - How does this study relate to this other study that is missing in their related works part: Detecting the ultra low dimensionality of real networks P Almagro, M Boguñá, MÁ Serrano Nature communications 13 (1), 6096?
>
> The work “Detecting the ultra low dimensionality of real networks” (P. Almagro, M. Boguñá, M. Á. Serrano, Nature Communications 13, 6096) does not provide a framework for exact network reconstruction or learning the dimensionality required for perfect reconstruction. Instead, their approach focuses on inferring network dimensionality by analyzing the relationship between the proportions of triangles, squares, and pentagons in a network, using synthetic networks with known dimensionality for training. Specifically, they generate various synthetic networks and employ a k-NN classifier, where k is selected based on the value that best classifies networks according to their ground-truth dimensionality. This method differs fundamentally from our work, which focuses on learning exact network embeddings that enable perfect reconstruction. In contrast, their approach leverages topological metrics as predictors of dimensionality but does not produce exact network embeddings or determine the precise dimensionality for embedding.
>
>
>
> - When the Authors mention in the limitation section the embedding using hyperbolic geometry, they refer to (Nickel & Kiela, 2017; 2018). It is not clear to me why they do not refer to:
>
> Thank you for pointing out these references. We have incorporated the suggested works into the related works section and discussed their relevance. Specifically, we added references to Krioukov et al. (2009), Boguñá et al. (2010), Papadopoulos et al. (2015), and the model-free approaches by Thomas et al. (2016, 2017). These contributions provide an important context for embedding in hyperbolic spaces.

---

> > ### Comment · Reviewer_odRC · 2024-12-03
> >
> > I thank the Authors for the effort to reply to my questions: I raised the final rating to 6: marginally above the acceptance threshold.

---

> > > ### Author Response · Authors · 2024-12-03
> > > **Thank you!**
> > >
> > > We once again sincerely thank the reviewer for their constructive feedback and valuable comments. We also appreciate their consideration in revising their score after reviewing our rebuttal responses.

---

### Author Response · Authors · 2024-11-22
**Updated paper and rebuttal response by authors**

We thank the reviewers for their time and thoughtful feedback, which have greatly helped us improve the clarity and experimental setup of our manuscript. In the revised version, all changes and updates are highlighted in blue for easy reference. We are happy to address any further questions or provide additional clarifications.

---

### Meta-Review · Area_Chair_T2s9 · 2024-12-20

**Metareview:**

Most reviewers were weakly positive on this paper and thus I am recommending acceptance. The paper gives a more scalable approach for finding exact low-dimensional embeddings of large networks, focusing on Euclidean embeddings. Using this approach they show that some real-world networks have surprisingly low exact embedding dimensions, extending and strengthening previous findings of Chanpuriya et al. A major weakness is that it's not clear what the upshot is -- the finding certainly seems interesting but what should we do with it? What implications does this have on graph ML methods or practice? In any case, reviewers found the paper sufficiently interesting as is to warrant acceptance.

I will note that some of the theoretical results in the paper seem weak/misrepresented. In particular, the claim in the abstract/intro that the paper affirmatively answers "Can metric model formulations provably provide lower dimensional representations that LPCA?" is confusing. It is clear that inner product and Euclidean norm embeddings are roughly the same due to the fact that ||x-y||^2 = ||x||^2 + ||y||^2 + 2<x,y>. In fact, this is confirmed by the paper's own theory: in Theorem 2.1 the authors prove that the two embedding approaches differ by at most an additive factor 2 in the embedding dimension (due to the rank-2 difference between a Gram matrix and a squared distance matrix. This is not a significant factor, and I would argue that it has essentially no relevance to our theoretical understanding of exact embeddability of real-world graphs. It does not deserve mention the abstract in my opinion or being highlighted as one of the main contributions of the paper.

The main takeaway from the paper should be that exact Euclidean embeddings are easier to find in practice on large networks, not that they have better representation capacity. This should be clarified.

**Additional Comments On Reviewer Discussion:**

Some additional experiments on synthetic low-dimensional geometric graphs were added, and some reviewer questions answered. Overall the rebuttal period did not lead to significant changes in the assessment of the paper.

---

### Decision · Program_Chairs · 2025-01-22

Accept (Poster)